# WordScape: a Pipeline to extract multilingual, visually rich Documents with Layout Annotations from Web Crawl Data

**Maurice Weber**[*]
ETH Zurich

**Carlo Siebenschuh**[*]
University of Chicago

**Rory M. Butler**[*]
University of Chicago

**Anton Alexandrov**
ETH Zurich,
INSAIT, Sofia University

**Valdemar R. Thanner**
ETH Zurich

**Georgios Tsolakis**
ETH Zurich

**Haris Jabbar**
TU darmstadt

**Ian Foster**
Argonne National Laboratory,
University of Chicago

**Bo Li**
UIUC

**Rick Stevens**
Argonne National Laboratory,
University of Chicago

**Ce Zhang**
ETH Zurich

{maurice.weber,ce.zhang}@inf.ethz.ch; {siebenschuh,rorymb}@uchicago.edu;
{aalexandrov, thannerv, gtsolakis}@student.ethz.ch;
harisjabbar@gmail.com; {stevens,foster}@anl.gov; lbo@illinois.edu

## Abstract

We introduce WordScape, a novel pipeline for the creation of cross-disciplinary, multilingual corpora comprising millions of pages with annotations for document layout detection. Relating visual and textual items on document pages has gained further significance with the advent of multimodal models. Various approaches proved effective for visual question answering or layout segmentation. However, the interplay of text, tables, and visuals remains challenging for a variety of document understanding tasks. In particular, many models fail to generalize well to diverse domains and new languages due to insufficient availability of training data. WordScape addresses these limitations. Our automatic annotation pipeline parses the Open XML structure of Word documents obtained from the web, jointly providing layout-annotated document images and their textual representations. In turn, WordScape offers unique properties as it (1) leverages the ubiquity of the Word file format on the internet, (2) is readily accessible through the Common Crawl web corpus, (3) is adaptive to domain-specific documents, and (4) offers culturally and linguistically diverse document pages with natural semantic structure and high-quality text. Together with the pipeline, we will additionally release 9.5M urls to word documents which can be processed using WordScape to create a dataset of over 40M pages. Finally, we investigate the quality of text and layout annotations extracted by WordScape, assess the impact on document understanding benchmarks, and demonstrate that manual labeling costs can be substantially reduced.

---

[*]The first three authors contributed equally.

37th Conference on Neural Information Processing Systems (NeurIPS 2023) Track on Datasets and Benchmarks.

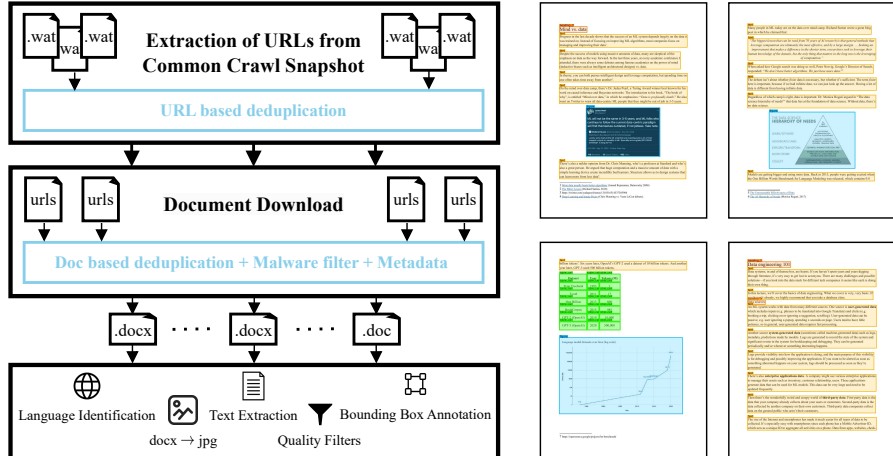

Figure 1: Overview over the WordScape pipeline for processing a single Common Crawl snapshot. First, we extract and deduplicate all URLs from .wat files that point to Word documents. We then download the documents, apply malware filters, metadata extraction and deduplicate based on content. In the third step, we convert downloaded documents to page images, extract text, run our bounding box annotation algorithm, identify the dominant language and apply quality filters. The right side of the figure shows an example document with bounding box annotations.

# 1 Introduction

There is an abundance of digital, semi-structured data contained in visually rich documents such as PDFs or MS Word documents. However, while this information is easily understood by humans, its semi-structured nature makes its analysis by automated data processing engines difficult. This difficulty stems, to a large extent, from the diversity of how information in visually rich documents is organized, across at least the three axes of culture, language, and industry. Therefore, effective use of such data often necessitates a costly and labour-intensive process of manual information extraction. Existing techniques in many automated document understanding tasks are either based on conventional rule-based or machine learning (ML) approaches, relying on hand-crafted features, or on more promising deep learning approaches which are trained on large amounts of data. However, to date, both approaches often fail to generalize well due to a lack of compatible formats, or due to insufficient diversity in existing training datasets, especially for low-resource languages.

At the same time, we have witnessed tremendous progress in the area of natural language processing (NLP) and computer vision applications, where pre-trained models [8, 25, 31, 24, 27] have enabled researchers and practitioners to build useful applications by fine-tuning these models on specific downstream tasks. This progress has, to a large extent, been driven by leveraging large quantities of high-quality data extracted from the web.

In contrast to these techniques, the task of document understanding is an inherently multimodal problem, requiring models to understand text, visual, and layout features and to model their relations [13, 22, 17]. As a consequence, next to advances in model design, considerable efforts have gone into creating datasets that combine these different modalities. These datasets can be grouped into two categories: on the one hand, there are human-labeled datasets like DocLayNet [23] for document layout analysis, FUNSD [14] for form understanding or the RVL-CDIP document classification dataset [12]. While this approach generates high-quality labels, it naturally limits the size to a few hundred thousand samples due to the restrictive costs of human-annotated labels. Alternatively, the automatic generation of ground truth labels has been leveraged to create datasets like PubLayNet [35], DocBank [19] and arXivdocs-weak [26]. While these datasets are larger in size, they are typically sourced from the scientific domain and are thus mainly in the English language and lack the diversity needed to reflect the true distribution of documents prevalent in practice and across industries, cultures, and societies.

Word documents are among the most widely used types of documents. While rendered Word documents appear as semi-structured pdfs, the source code comprising Word documents consists of

highly structured XML files in the Open XML format, containing valuable information like reading order, the structure of tables and style information. Furthermore, Word documents are often used in more formal contexts and for professional writing such as academic papers, reports, business documents or official correspondence. This gives rise to the hypothesis that text found in such documents generally has higher quality than text on web pages like forums, social media updates or user-generated content. Here, we present WordScape, a pipeline that enables the automatic sourcing and annotation of diverse, multilingual, visually rich documents at scale, enabling researchers and practitioners to curate multimodal document understanding datasets. Similar to large-scale NLP dataset creation pipelines like CCNet [33], we use the Common Crawl web corpus [2] as our primary source of documents. We parse Common Crawl to collect urls pointing to MS Word documents embedded in websites, then parse the Open XML structure of these documents to extract text and identify the location and category of visual semantic entities like section headings, tables and figures on the rendered page images. To date, we extracted a total of 9.5M urls to Word files which we will publish and which can be processed using the WordScape pipeline to build a corpus of roughly 40M pages. In summary, we make the following contributions:

- We present a novel pipeline to automatically extract and process millions of MS Word documents from the web, and open-source the codebase [3]
- We introduce a novel bounding box labeling algorithm based on the Open XML representation of MS Word documents.
- We provide a detailed analysis of the size, quality and distribution of datasets created using WordScape.
- We validate one created dataset on various layout analysis benchmarks and find that WordScape annotations can substantially reduce manual labeling efforts.
- We will release 9.5M urls to word documents that we have collected from Common Crawl. These can be processed using WordScape to create a dataset of over 40M pages.

The remainder of this paper is organized as follows: In Section 2 we discuss related work. In Section 3 we present our data creation pipeline and dataset metrics are presented in Sections 4 and 5. We validate our dataset on downstream benchmarks in Section 6, and finally conclude in Section 7.

## 2 Related Work

With the proliferation of pre-trained deep learning models, there has been a growing focus on high-quality, web-scale training datasets, both in the domains of natural language processing and computer vision. In the NLP domain, notable advances include the CCNet pipeline, C4 [25, 34], OpenWebText [11], Pile v1 [9], S2ORC [20], Pythia [2], the RedPajama dataset [6], or the Refined Web dataset [21]. Similar to these datasets and pipelines, this work aims to leverage data that is publicly available on the web to build a large-scale, diverse, and multilingual data corpus. However, here we focus on visual document understanding, an inherently multimodal domain, where next to text, layout and visual features of documents are also a valuable source of data for downstream models.

Perhaps more similar to this work are multimodal web-scale datasets and pipelines like the Laion-400M and Laion-5B datasets [30, 29], or the multimodal mmC4 [36]. In contrast to these works, we focus on visually rich documents with layout annotations, rather than *(image, caption)*-pairs or text interleaved with images. There are multiple datasets in the visual document understanding domain such as the manually annotated DocLayNet [23] for document layout analysis, FUNSD [14] for form understanding or the RVL-CDIP document classification dataset [12]. Other notable datasets include the automatic, weakly labelled PubLayNet [35], DocBank [19], TableBank [18] and arXivdocs-weak [26] datasets. In a similar manner to our approach, the LayoutReader dataset [32] leverages the Open XML format of Word documents to construct a multimodal dataset of visually rich documents together with the text in reading order. However, LayoutReader does not include object detection labels for semantic entities and is released as a fixed dataset, rather than as a dataset creation pipeline. In Table 1, we show a detailed comparison between WordScape and other datasets and pipelines.

---

[2]https://commoncrawl.org/
[3]https://github.com/DS3Lab/WordScape

Table 1: Comparison with existing document layout datasets. (1) WordScape is released as a public pipeline together with 9.5M document urls; (2) 28 top-level categories detected via hierarchical topic modelling; (3) languages detected with fastText on a single common crawl snapshot [16, 15].

| Dataset | Pages | Classes | Annotation | Format | Document Types | Languages | Source |
|---|---|---|---|---|---|---|---|
| PubLayNet [35] | 360k | 5 | Automatic | Digital | Scientific articles | English | PubMed Central |
| DocBank [19] | 500k | 13 | Automatic | Digital | Scientific Articles | English | arXiv |
| arXivdocs-weak [26] | 127,472 | 23 | Automatic | Digital | Scientific Articles | English | arXiv |
| PRImA [1] | 305 | 10 | Automatic | Scans | Magazines, Technical Articles, Forms, Bank Statements, Advertisements | English | – |
| DocLayNet [23] | 80,863 | 11 | Manual | Digital | Financial Reports, Manuals Scientific Articles, Laws & Regulations, Patents, Government Tenders | English, German French, Japanese | – |
| $M^6$Doc [5] | 9,080 | 74 | Manual | Digital, Scan, Photographs | Scientific articles, Textbooks, Books, Test papers, Magazines, Newspapers, Notes | English, Chinese | Chinese People's daily, arXiv, VKontakte |
| WordScape (Ours) | 9.5M urls[(1)] | 30 | Automatic | digital | > 28 Categories[(2)] | > 136 languages[(2)] | Common Crawl |

# 3  Methodology

Our document processing pipeline builds on the Open XML structure of Word documents which contains valuable semantic information, and the hypothesis that the quality of text contained in such documents is generally higher than text found on HTML-based websites. As our primary source of data, we use the Common Crawl web corpus consisting of regular snapshots of the web with little overlap between different snapshots [4], dating back to 2013. On a high level, our pipeline consists of three core steps: we first parse a Common Crawl snapshot and extract all links that point to Word files (i.e. URLs that end in .docx or .doc). The second step is to send HTTP requests to these links and download the corresponding Word file. The final step consists of processing the document, resulting in a final multimodal dataset with page images, text contained on each page, and bounding box annotations for semantic entities like headings and tables on each page. An overview of the pipeline is shown in Figure 1. In this section, we present each step in more detail.

## 3.1  Parsing of Common Crawl

The first step in the WordScape pipeline is to extract urls that point to Word files from Common Crawl snapshots. Common Crawl provides data in raw (`warc`) format, UTF-8 encoded text (`wet`) and metadata (`wat`) files. Next to HTTP header data, each metadata file also contains a list of hyperlinks, from which we select all HTTP urls that end in .doc or .docx. Each `wat` file must be downloaded in its entirety to be correctly parsed. After the initial parsing of the `wat` files, the urls from each agent are merged, then deduplicated on a per-snapshot basis and finally deduplicated globally across all previously processed snapshots.

## 3.2  Document Download

In this step of the WordScape pipeline, we download the documents from the urls extracted from Common Crawl. This step outputs the downloaded Word source files, as well as metadata containing statistics on the quantity of successfully downloaded urls and failure or rejection reasons for each url. Failures relate mainly to benign HTTP errors. We reject a document when a response is successfully obtained but is not useful. A benign reason for rejection is either an unsuccessful HTTP response code (most commonly 403/404), an invalid URL, too many redirects/retries, no received HTTP response, an incorrect file format of the response, an incorrect content-type header, or internal hardware failure/cluster resource limitations. We furthermore reject potentially malicious documents by performing a check against OLE data structures [7] during download. As such we count any document that contains VBA code/macros, external relations, an OLE object pool, encryption, or flash embeddings.[5] Finally, we also reject excessively large files. The reason for this is that we aim to achieve a relatively even distribution of document pages, and to prevent out-of-memory errors.

---

[4] `https://commoncrawl.github.io/cc-crawl-statistics/plots/crawloverlap`

[5] While this is a conservative malware filter, we emphasize that motivated adversaries can in theory still engineer malicious documents. Our malware filter is thus not a security guarantee.

Upon a successful download, we save several metadata fields concerning the response and its analysis such as HTTP status, OLE information, as well as the file itself. In addition, the metadata includes a SHA-256 hash of the full response bytes: This allows us to perform a second global deduplication step against files which have identical content but are accessible under different urls, and to ensure that the document has not changed if it is downloaded a second time. This serves as a defense against potential dataset poisoning attacks, which have recently been shown to be practical in the context of web-scale datasets [4]. Finally, the temporary metadata files recorded by the agents are merged, deduplicated via bytehash, and written to a database. We present metrics on this metadata in Section 4.

### 3.3 Document Processing

The processing of Word documents consists of several steps, including language identification, bounding box annotation and text extraction. Here we describe each step in detail.

#### 3.3.1 Bounding Box Annotation

Similar to the approach from [19, 18], we use a colorization scheme to extract bounding boxes of semantic entities from a document page. In the first step, we parse the highly structured Open XML files of a Word document using the `python-docx`[6] library and custom XML parsing code to identify the categories of different elements in the document.

We identify such elements by one of two methods: If the Word user has either used a built-in style (such as a heading formatter), or the element is natively tagged in the XML file (such as for tables), we use this information to label the corresponding element. Clearly, this approach makes the assumption that using such a built-in functionality reflects the user's intent, which is not always the case. Nevertheless, we expect this methodology to be accurate in the majority of cases. If, on the other hand, no built-in indicator can be found for an element, we fall back to heuristics, such as the distribution of used fonts in the document indicating headings, or successive numbered or bulleted paragraphs indicating a succession of list items. This heuristics-based approach is generally more noisy compared to the method based on built-in XML properties.

Once the category of a document element has been determined, we color it using the Open XML formats highlighting, formatting and text coloring features, by directly editing the XML. The colored document is then rendered via LibreOffice [7], and each page is converted to an image. Colors on this image are then detected, providing bounding boxes for each different entity category. We provide more details on the annotation process in the Appendix.

#### 3.3.2 Text Extraction

We extract text from a document on two levels of granularity. First, we extract the full document text from the Open XML structure using `python-docx`. This document-level text is in reading order, due to the internal XML structure. Second, we extract the text from individual rendered pages using the `PDFPlumber`[8] package. This allows us to additionally extract word-level bounding boxes. It should be mentioned that the PDF-based extraction is less accurate due to the necessity to use heuristics when identifying and grouping characters into words. We discard any document that has less than a total of 200 characters. On the page level, we keep pages without any text as they might contain figures or other relevant entities without text.

#### 3.3.3 Language Identification

To identify the language of a document, we use the fastText language classifier [16, 15]. The classifier was trained on Tatoeba, Wikipedia and SETimes and can identify 176 languages using $n$-grams as features with the hierarchical softmax. We identify languages both on a document level, using the Open XML-based text, and on a page level using the PDF-based text.

---

[6] `https://github.com/python-openxml/python-docx`
[7] `https://www.libreoffice.org/`
[8] `https://github.com/jsvine/pdfplumber`

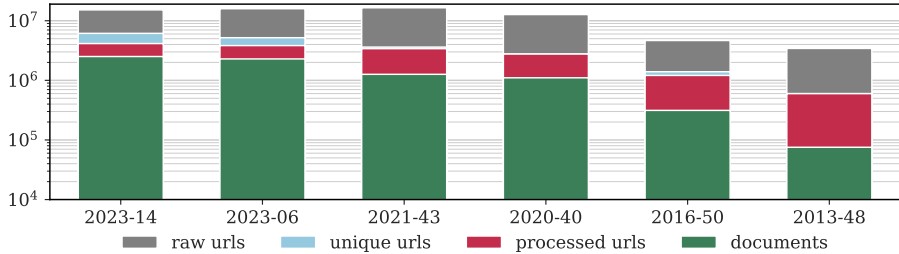

Figure 2: Number of urls and documents extracted from Common Crawl, for snapshots ranging from 2013 to 2023. Raw urls refers to the initial urls extracted from each snapshot, prior to any processing. The unique urls are globally deduplicated, starting from the most recent snapshot back to the oldest. Processed urls is the subset of urls to which an HTTP request was sent, and "documents" refers to successfully downloaded documents.

### 3.3.4 Dataset Filters

We provide several ways to filter a subset of the core dataset created by WordScape, based on metadata collected during the annotation process. First, we implement a quality filter based on the perplexity of the document text for models trained on Wikipedia, as well as a annotation reliability score to assess bounding box annotations. The latter metric captures the proportion of entities annotated using built-in or XML patterns vs. heuristic-based annotations, as the former are generally more reliable. We present more details on the perplexity distribution in Section 5 and details on the annotation quality score in the Appendix. In addition, we collect metadata for each document and each page, allowing the creation of subsets with different requirements such as the number of tables and other entities, or the language of the resulting dataset.

## 4 Pipeline Statistics

In this section, we present statistics on running the WordScape pipeline. Specifically, we investigate the number of links pointing to MS Word files in a single common crawl snapshot, as well as the reasons that downloads failed, or were otherwise rejected/failed during the annotation process. We provide details on the resources used to run the WordScape pipeline in the Appendix.

**Common Crawl Parsing** To estimate the number of documents that can potentially be obtained using the WordScape pipeline, we parsed 6 individual Common Crawl snapshots ranging from 2013 up to the March/April 2023 snapshot. We found that there are substantial duplicated Word file urls in each snapshot: per-snapshot deduplication removes $60 - 80\%$ of available urls. However, we found little overlap *between* snapshots, mirroring the fact that there is generally little overlap between the websites visited by Common Crawl. Furthermore, we noticed that the number of valid urls, i.e. where a document can be successfully downloaded, decreases substantially for older crawls: Out of all the urls visited for the 2013 snapshot, only $12.5\%$ could be successfully downloaded, compared to $60.6\%$ of urls from the most recent 2023 snapshot. This is to be expected as older urls are more likely to be inaccessible than newer ones. These observations are illustrated in figure 2.

**Document Download** Out of a total of $5,807,634$ requests to urls from the November/December snapshot, we found that $2,441,972$ $(42.1\%)$ received a 200 return code and could thus be further processed. Out of the successful responses, there were $364,648$ $(14.9\%)$ instances where the content-type header did not match a Word document and was thus rejected. Another $172,772$ $(7.1\%)$ documents were rejected because they did not pass our OLETools malware filter. This resulted in a total of $1,904,552$ Word documents that could be successfully downloaded using the WordScape pipeline. We emphasize that we ran the requests at the beginning of March 2023, i.e. roughly three months after the snapshot was published. It is likely that at a later point in time, the number of responsive urls will be lower, leading to less documents. We provide further details on reasons for rejected downloads in the Appendix.

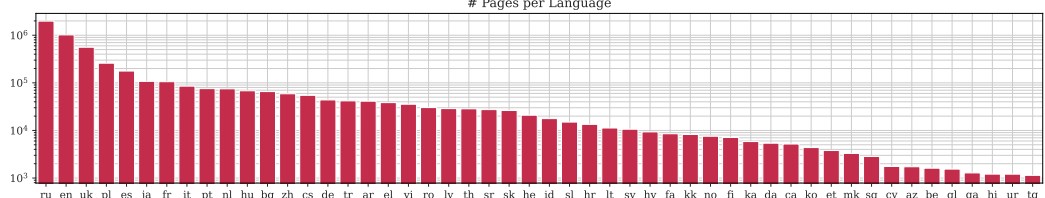

Figure 3: Number of pages per language, produced by the WordScape pipeline run on $1.25M$ Word documents extracted from the November/December 2022 Common Crawl snapshot.

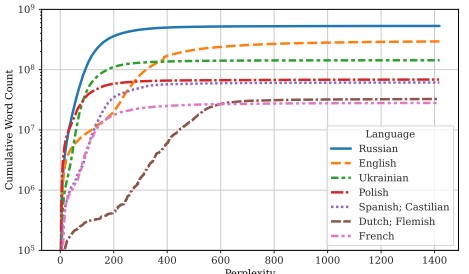 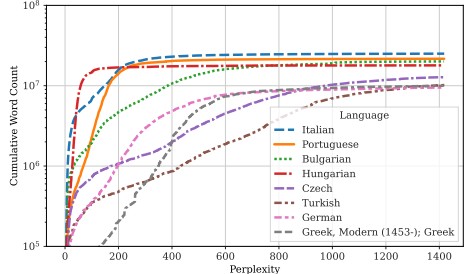

Figure 4: Number of words in any given language subset, as a function of perplexity threshold. The figure shows the number of words with perplexity smaller or equal to the value on the x-axis, for the seven languages with the highest (left) and lowest (right) number of words. A word is defined as a whitespace-delimited sequence of characters with punctuation removed.

**Document Processing**    To further investigate how the WordScape pipeline performs, we ran the document processing step of the pipeline on $1,251,383$ of the successfully downloaded documents from the November/December 2022 snapshot. Out of these, for $248,918$ ($\sim 19.9\%$) of the documents, the annotation process was either rejected or failed. The majority of the failures stem from the files being invalid Zip files, namely in $15.4\%$ of all processed documents. Another $37.5k$ ($\sim 3\%$) of the documents were rejected because they contained less than 200 characters. We furthermore rejected all documents whose uncompressed file size was more than 20 times the compressed size as this indicates a potentially malicious zip bomb. This process resulted in $1,002,465$ annotated documents, or $5,481,455$ pages, including the document text and object detection bounding boxes.

## 5    Dataset Statistics

**Language Distribution**    In the $5.5M$ annotated document pages, we found a total of 136 distinct languages, identified with fastText [16, 15]. The page counts per language is highly skewed towards high-resource languages like Russian (2M pages) and English (1M pages), as opposed to roughly 1k pages for Tajik and Urdu. Figure 3 shows the number of pages for the 50 most frequent languages.

**Perplexity Scores**    We use the perplexity of a language model trained on a target domain to measure the quality of the text extracted using WordScape. We follow the approach used in [33] and use their 5-gram Kneser Ney models and SentencePiece tokenizers trained on Wikipedia. In this context, a lower perplexity score indicates that the language is closer to the target domain and is thus expected to be of higher quality. Figure 4 shows the number of words with at most a certain perplexity value. We note that especially for Hungarian, Portuguese and Italian, the perplexity scores are relatively low, and a large part of the corpus can be retained even when aggressively filtering out documents with moderately high perplexity. We provide further figures that illustrate the perplexity distributions for more languages in the Appendix.

**Semantic Entity Distribution**    Semantic Entities like headings, tables and lists are the logical units that build up the structure of a document. Here we present statistics on the semantic entities that the WordScape pipeline annotates. This analysis is based on the $1.25M$ documents annotated from

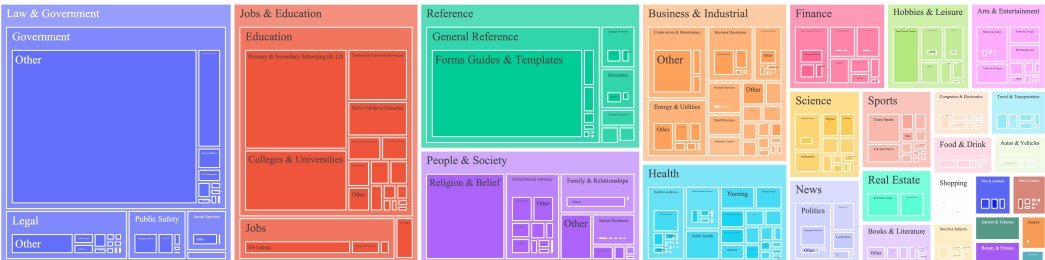

| | list | text | heading | form_field | table | figure | footer | header | title | toc | form_tag | quote | equation | footnote | bibliography | annotation |
|---|---|---|---|---|---|---|---|---|---|---|---|---|---|---|---|---|
| en | 26.884% | 32.131% | 18.617% | 6.791% | 4.621% | 3.264% | 3.859% | 2.595% | 0.654% | 0.158% | 0.168% | 0.069% | 0.065% | 0.033% | 0.071% | 0.021% |
| es | 25.438% | 30.817% | 17.096% | 5.063% | 5.082% | 5.701% | 3.338% | 6.402% | 0.744% | 0.152% | 0.058% | 0.033% | 0.031% | 0.030% | 0.003% | 0.014% |
| fr | 22.335% | 32.870% | 18.777% | 7.119% | 5.794% | 5.989% | 3.370% | 2.510% | 0.702% | 0.154% | 0.227% | 0.051% | 0.046% | 0.026% | 0.005% | 0.025% |
| nl | 22.065% | 40.949% | 15.798% | 3.777% | 2.838% | 3.173% | 2.727% | 7.387% | 0.499% | 0.037% | 0.080% | 0.476% | 0.083% | 0.093% | 0.001% | 0.017% |
| pl | 37.811% | 21.402% | 14.113% | 13.717% | 4.490% | 3.215% | 2.914% | 1.695% | 0.479% | 0.039% | 0.039% | 0.015% | 0.007% | 0.043% | 0.000% | 0.019% |
| ru | 40.770% | 28.777% | 11.660% | 7.673% | 6.993% | 1.226% | 1.142% | 1.341% | 0.298% | 0.047% | 0.004% | 0.023% | 0.027% | 0.014% | 0.000% | 0.005% |
| uk | 33.043% | 33.565% | 18.179% | 4.219% | 6.443% | 2.408% | 0.405% | 1.008% | 0.661% | 0.031% | 0.004% | 0.013% | 0.009% | 0.004% | 0.000% | 0.006% |
| Total | 31.970% | 30.349% | 15.666% | 7.981% | 6.035% | 2.663% | 2.356% | 2.152% | 0.549% | 0.079% | 0.062% | 0.044% | 0.037% | 0.029% | 0.016% | 0.013% |

Figure 5: Proportion of layout element categories for the seven most frequent languages.

Figure 6: Hierarchy of topics detected in the WordScape.

the November/December 2022 snapshot, containing a total of 173M entity bounding boxes. The most frequently appearing category are table cells (86M individual cells), making up roughly 50% of the entities. Table columns and rows also appear frequently, comprising another 20% of all entity bounding boxes. The next most frequent category are list items, of which we found 15M (8.9%). Further frequent categories are plain text (14.8M) and headings (7.6M). We found that, generally, the entity distribution is highly imbalanced; however, when excluding the individual table elements, the distribution flattens significantly.

In Figure 5 we show the semantic entity distribution, grouped by languages and where we have excluded table elements and merged the different heading levels into a single category. We see that the imbalanced nature of the categories persists across all languages considered. However, there are differences between languages in regard to the extent to which the classes are imbalanced (e.g. Russian documents are more imbalanced than French documents). To whether the pairwise occurrence of layout elements is correlated, we compute Spearman's rank correlation coefficient for pairs of layout elements in Figure 10 in the Appendix. We found that the elements are generally weakly correlated, except for (text, heading), (list, heading), (list, text), (form_field, text) and (footer, header). Further details on the semantic entities are in the Appendix.

**Topic Modeling**    Since Common Crawl snapshots cover websites across multiple, unfiltered domains, the Word documents extracted via WordScape can potentially themselves also cover a wide range of topics. To get a better understanding of the topic distribution in WordScape, we ran the hierarchical topic modelling classifier available in the Google Cloud NLP API [9] over a 25k sample of WordScape documents in 11 languages support by the API (ru, en, it, ja, es, nl, zh, pt, ko, fr, de). This allows for a fine-grained analysis giving both a high-level overview of the topics in the dataset, and also exposes more low-level details of sub-classifications. The top categories we found are "/Law & Government/Government/Other" (15.3%), "/Reference/General Reference/Forms Guides & Templates" (9.0%), "/Jobs & Education/Education/Primary & Secondary Schooling (K-12)" (5.6%), "/People & Society/Religion & Belief" (3.5%), and "/Jobs & Education/Education/Colleges & Universities" (3.4%). The full hierarchy is shown in Figure 6. We furthermore split the analysis across languages that where we found significant differences in topic distributions. While the most frequent top-level category in both Russian and Portuguese is "Law & Government", accounting for ∼ 37% of documents, this category occurs relatively infrequently in other languages (< 13%). We provide a more fine-grained overview over the language specific topic distributions in the Appendix.

---

[9] https://cloud.google.com/natural-language

# 6 Training Object Detection Models on WordScape

To further assess the quality of the bounding box annotations extracted by WordScape, we conduct experiments on three different datasets. We measure the utility of the annotations by first training a base model on WordScape annotations, then finetuning the model on a target benchmark. We attempt to show that, by leveraging the automatically annotated labels produced by WordScape, we can reduce the (manual) labeling cost on the target domain while still maintaining the original performance.

**Text Detection on FUNSD**  We first consider the word-level text detection task on the FUNSD dataset [14]. This dataset is a subset of the RVL-CDIP [12] dataset and comprises 199 manually annotated, scanned forms. We trained a Faster R-CNN [28] network on 0 to 100k pages, annotated with word-level bounding boxes. In the second step, we finetuned the resulting model on 25 - 149 samples of the FUNSD dataset. In Table 2 we report the F1 score with IoU thresh-

Table 2: Text detection F1 @IoU 0.5 for Faster R-CNN on FUNSD. $N_p$ are the WordScape samples; $N_f$ the FUNSD samples.

|  | $N_f = 25$ | $N_f = 50$ | $N_f = 100$ | $N_f = 149$ |
|---|---|---|---|---|
| $N_p = 0$ | 0.621 | 0.690 | 0.723 | 0.772 |
| $N_p = 10k$ | 0.840 | 0.840 | 0.823 | 0.861 |
| $N_p = 50k$ | 0.868 | **0.870** | **0.857** | 0.869 |
| $N_p = 100k$ | **0.872** | 0.869 | 0.850 | **0.882** |

old 0.5. We can see that using only 10k WordScape samples and 25 finetuning samples substantially surpasses the text detection accuracy of the model trained on the full FUNSD dataset. By using WordScape annotations we can thus decrease the labeling cost 6-fold.

**Table Detection on ICDAR 2019 cT-DaR**  Here we present results on the ICDAR 2019 cTDaR table detection task [10]. We use the modern tables subset, the domain of which is closer to the WordScape domain, compared to the archival subset. It includes 600 training images and 240 test images. We compare mAP @ IoU [0.50:0.95] for the Ultralytics YOLOv8m[10] model on the test set with and without any

Table 3: Table detection mAP @ IoU [0.50:0.95] for YOLOv8m on ICDAR 2019 cTDaR. $N_p$ are the WordScape samples, $N_f$ the cTDaR samples.

|  | $N_f = 75$ | $N_f = 150$ | $N_f = 300$ | $N_f = 600$ |
|---|---|---|---|---|
| $N_p = 0$ | $0.869 \pm 0.008$ | $0.888 \pm 0.011$ | $0.949 \pm 0.006$ | $0.974 \pm 0.003$ |
| $N_p = 1.25k$ | $0.906 \pm 0.012$ | $0.912 \pm 0.011$ | $0.951 \pm 0.005$ | $0.972 \pm 0.003$ |
| $N_p = 2.5k$ | $0.914 \pm 0.009$ | $0.929 \pm 0.008$ | $0.960 \pm 0.004$ | $0.974 \pm 0.003$ |
| $N_p = 5k$ | $\mathbf{0.924} \pm 0.007$ | $0.924 \pm 0.011$ | $0.956 \pm 0.005$ | $0.974 \pm 0.003$ |
| $N_p = 10k$ | $0.919 \pm 0.006$ | $\mathbf{0.931} \pm 0.010$ | $\mathbf{0.961} \pm 0.005$ | $\mathbf{0.975} \pm 0.003$ |

training on WordScape table documents. We resize images to $640 \times 640$ resolution and train 4 models with different training set sizes for 200 epochs using SGD, 0.01 learning rate, batch size of 16, 0.937 momentum and $5e - 4$ weight decay. We then finetune each model on 4 different subset sizes of cTDaR using AdamW with $5e - 4$ learning rate for 200 epochs, or until no improvement is observed for more than 30 epochs. We see that pre-training on WordScape improves results particularly in the low-resource regime.

**Layout Analysis on DocLayNet**  DocLayNet [23] is one of the largest human-annotated document layout segmentation datasets, containing over 80k pages from a variety of document sources. We train a YOLOv5 object detection model on 200k images obtained via WordScape, and then fine tune the model on subsets of the Do-cLaynet training split, varying the fine

Table 4: Document Layout Analysis mAP @ IoU [0.50:0.95] for YOLOv5 on DocLayNet with different pretraining datasets. $N_f$ is the number of finetuning samples.

|  | $N_f = 1k$ | $N_f = 5k$ | $N_f = 20k$ | $N_f = 69k$ |
|---|---|---|---|---|
| Random Initialization | 0.299 | 0.553 | 0.727 | 0.753 |
| PubLayNet (200k) | 0.467 | 0.659 | 0.720 | 0.745 |
| WordScape (200k) | **0.508** | **0.679** | **0.734** | **0.755** |

tuning dataset sizes from $1k$ to the full $69k$. The results are shown in Table 4, where we see that pre-training on WordScape leads to consistent performance improvements compared to (1) using random weights for intialization, and (2) pretraining with the same number of samples on the PubLayNet [35] dataset. This is particularly pronounced when less human labelled data is available.

**Handcrafted Scientific Dataset**  WordScape's versatility arises from enabling access to multilingual document pages with rich category structure. However, assessing its quality requires an

---

[10]https://github.com/ultralytics/ultralytics

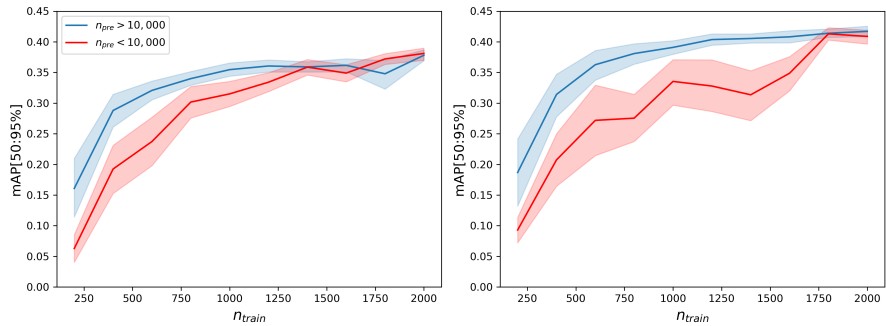

Figure 7: Layout analsysis mAP @ IoU [0.50:0.95] on the scientific paper dataset with YOLOv8 (left) and DETR (right) for varying WordScape sample sizes. The figures show mean values for smaller (2.5k, 5k, 7.5k, red) and larger (15k, 20k and 25k, blue) pretraining sizes. The confidence bands arise as estimates of the quartiles $q_{0.25}$ and $q_{0.75}$.

equally refined dataset for downstream tasks. The growing interest in layout detection as a precursor to multimodal document understanding makes scientific literature a particularly viable candidate.

We compiled a dataset of diverse scientific content of $N_f = 2,500$ pages from eight scientific domains (biology, chemistry, physics, mathematics, engineering, computer science, economics, and medicine) that span 120 subdomains from abstract algebra to zoology. More than 41,000 instances were annotated by humans. In total, there are 31 categories that can be grouped into metainformation (e.g. title), text body (e,g, paragraph), code (e.g. pseudocode), mathematical content (e.g. equation), and visual assets (e.g .table). Categories relating to text are further split by indicating the presence of LaTeX-formatted symbols. In addition, categories are present that relate captions to the respective figure or table. Finetuning a model on an information-dense, annotation-rich scientific corpus is a formidable challenge for a variety of reasons. It is particularly daunting for a model pre-trained on Wordscape, however, due to the (1) multi-disciplinary, information-dense scientific content, (2) the extensive, hierarchical class labels, (3) the high-resolution images stored as PNG rather than JPG, (4) the human-made annotations and (5) the distributional shift from multi-lingual word documents to English-only PDFs.

DETR [3] and the current iteration of YOLO represent state-of-the-art choices in terms of accuracy and latency, respectively. In our experiment, both models are pre-trained on pages stemming from WordScape with sample sizes ranging from 1000 to 100K. Subsequently, the models are finetuned on our handcrafted scientific dataset for up to 2,000 pages. The empirical results are shown in figure 7 and indicate that pre-training on WordScape significantly reduces the need for downstream data.

## 7 Discussion

In this paper, we present a pipeline to create curated datasets consisting of high-quality, multilingual and diverse, visually rich documents with layout annotations. The pipeline is scalable to millions of pages and contains high-quality text, both for high- and low-resource languages. WordScape is the first pipeline that enables the creation of training datasets for large-scale multimodal document understanding models that fuse text, visual and layout features. As the main limitation of the pipeline, we identify the reliability of the bounding box annotations for certain semantic entities like headings, as they rely to some extent on the assumption that formatting correlates reasonably well with user intent. In the future, we wish to explore more characteristics of the resulting dataset such as the amount of toxic or offensive content, and train large-scale document understanding models that make full use of both text and image modalities in multiple languages. We are also excited to see how this dataset can further enhance existing web-scale multimodal and NLP datasets.

## Acknowledgments and Disclosure of Funding

This work is partially supported by the National Science Foundation under grant No. 1910100, No. 2046726, No. 2229876, and Alfred P. Sloan Fellowship. CZ and the DS3Lab gratefully acknowledge the support from the Swiss State Secretariat for Education, Research and Innovation (SERI) under contract number MB22.00036 (for European Research Council (ERC) Starting Grant TRIDENT 101042665), the Swiss National Science Foundation (Project Number 200021 184628, and 197485), Innosuisse/SNF BRIDGE Discovery (Project Number 40B2-0 187132), European Union Horizon 2020 Research and Innovation Programme (DAPHNE, 957407), Botnar Research Centre for Child Health, Swiss Data Science Center, Alibaba, Cisco, eBay, Google Focused Research Awards, Kuaishou Inc., Oracle Labs, Zurich Insurance, and the Department of Computer Science at ETH Zurich. IF and RS acknowledge support from the U.S. Department of Energy under Contract DE-AC02-06CH11357.

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

# A  Appendix

## A.1  Details on Semantic Entity Annotation

One central part of the WordScape pipeline is the segmentation of page images into different semantic entities. As discussed in the main part of the paper, we segment the pages as follows:

1. Identify the classes of different elements in the documents by parsing the Open XML structure of Word documents.

2. Edit the XML tags such that the elements appear in a specific color on the rendered page images. We first eliminate any highlighting, and then change the color of the font as well as the background, ensuring that the change in color does not affect the spatial composition of elements on the pages.

3. Match the colors on rendered pages using OpenCV [11] to get bounding box annotations for each category.

In this way, we annotate the following set of semantic entities:

```
Title, Heading Level 1, Heading Level 2, Heading Level 3, Heading Level 4,
Heading Level 5, Heading Level 6, Heading Level 7, Heading Level 8, Heading
Level 9, Plain Text, List Item, Header, Footer, Table Header, Table Header
Cell, Table, Table Cell, Table of Contents, Bibliography, Quote, Equation,
Figure, Table Caption, Footnote, Annotation, Form Field, Form Tag, Table
Row, Table Column.
```

As discussed previously, we have several ways to identify the category of an element. Here we discuss each technique in more detail.

**Built-in Styles**  Word has a number of built-in formatting styles which users can apply to achieve a well structured and formatted document. This includes styles for titles, headings with different levels, plain text, list items, footnotes and others. We take the usage of a given builtin style as a signal to identify the entity category of the element in the Open XML files and colorize the element accordingly.

**Open XML tag**  Some elements cannot be readily identified via builtin styles, but rather using specific Open XML tags, accessible via the `python-docx` library. In our implementation we use this methodology to identify the following categories: `header`, `footer`, `text box`, `table`, `table cell`, `built-in table of content`, `built-in forms`, `figures`. Note that we relabel text boxes as plain text. In the context of XML tag identification, we refer to table of content and forms as "built-in" as opposed to heuristically matched toc and forms. The colorization of figures is implemented by replacing the image files in the Word zip files with an image of the same shape and format (e.g., jpg or png encoded), but filled with the color corresponding to the figure entity.

**Heuristics**  In the case that we cannot identify an element using built-in styles or XML indicators, we fall back to heuristics. These fall broadly under three categories:

Firstly, we compare the elements styling to the styling of built-in elements. The following user action serves as an example: A user creates a heading using the built-in heading feature and then styles that built-in element, and later copies and pastes the built-in heading to a different location in the document, then edits its text content. The first element would be detected as a built-in heading, but the second element would not. The second copy-pasted element possesses no built-in style name indicator; however, it possesses identical applied styling (font size, boldness, underlines etc.) to the known built-in element, and should therefore receive the same classification. We perform a second pass on each document after built-in elements have been found in order to classify any non-builtin elements with identical applied styling.

Second, we use content-aware heuristics specific to individual element types. For example, a paragraph in which every line break is immediately followed by a number or special bullet character

---

[11]https://github.com/opencv/opencv-python

is classified as a list, and text segments above a certain length consisting only of underscores are classified as form fields.

The last and most rudimentary heuristic deals only with distinguishing plain text elements from title and heading elements; we choose this approach as our last fallback because heading elements are crucial in outlining document structure. We rank elements according to font size, boldness and underlining, and use this information to create a hierarchy among text elements which matches Words built-in heading feature. For example, if a document consists of elements with font sizes 20, 16 and 12, this approach would classify elements of size 20 as heading 1, size 16 as heading 2, and size 12 as plain text. As this is the last heuristic fallback, and therefore the least reliable, we employ various checks and restrictions on these classifications. For example, We configure a maximum length for headings classified this way, only rank font sizes as indicating a heading if they are larger than the most commonly appearing font size, and only classify document titles if their styling is unique and of the largest font size. Finally, if an element does not match any heuristic, we simply label it as plain text.

**Table Rows and Columns** We find table rows and columns by post-processing the bounding boxes for table cells. Specifically, we divide a table into the grid corresponding to the finest granularity; i.e., the smallest cell height for rows, and the smallest cell width for columns. After ordering the cells from top to bottom, a row at vertical position $y$ is then defined as the sequence of all cells that vertically cover the cell with smallest height and starting at position $y$, ordered from left to right. Columns are determined analogously. Note that in this way we account for merged cells, i.e., the same (merged) cell can appear in multiple rows / columns. This is similar to the internal representation of tables in the Open XML standard.

## A.2 Quality Filters

In WordScape, we calculate several quality indicators based on which subsets of the output can be filtered.

**Preliminary Filters** As a preliminary step, during processing, we discard any document that has less than 200 characters in it. In addition, we discard documents with more than 150 pages or whose absolute (compressed) file size is larger than 10MB in order to maintain document diversity. We also discard documents whose uncompressed size is more than 20 times its compressed size, or documents that contain excessively large images (> 22.4M pixels).

**Text based characteristics** For each document, we collect the number of characters, the number of words [12], the number of alphabetical characters, the number of numerical characters, the alphanumerical proportion, and the ratio of of alphabetical to numerical characters. In addition, we provide utilities to compute the perplexity of Wikipedia trained language models as used in the `CCNet` pipeline [33]. Finally, each document is classified according to its dominant language using the FastText classifier [16, 15]. Here we also include the classification confidence in order to maintain the possibility to filter out low-confidence documents.

**Bounding box annotations** The WordScape bounding box annotations stem from either built-in and Open XML related sources, or from heuristics. We found that the former source is generally more reliable as the user has less degrees of freedom (e.g., tables) and choosing a particular builtin style is a conscious action which we argue is a strong signal to their intention. Heuristics on the other hand are based on relative font sizes and special characters and thus provide a much weaker signal. To capture these different sources of bounding box annotation, we compute an annotation reliability metric, which as a weighted average over the proportion of the number of characters of reliably (i.e., builtin or XML tag based) annotated entities against the number of characters for heuristic-based annotations. Formally, we have the following score

$$R = \sum_{i=1}^{N} \gamma_i r_i, \qquad \gamma_i = \frac{c_i}{\sum_{i=1}^{N} c_i}, \quad r_i = \frac{b_i}{b_i + h_i} \tag{1}$$

---

[12]We define a word as a white-space delimited sequence of characters with punctuation removed.

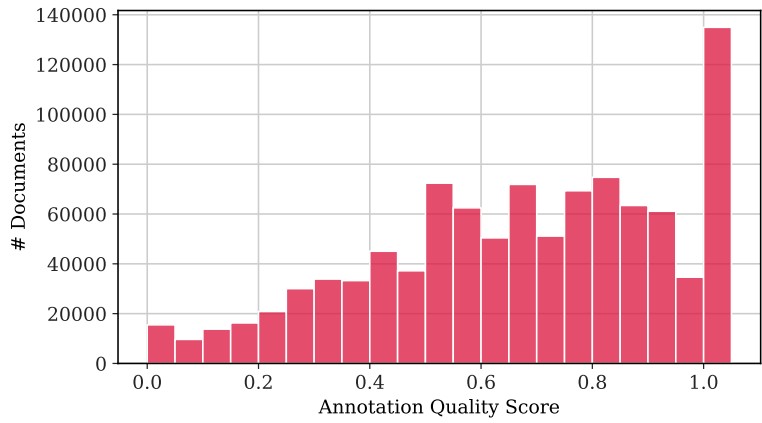

Figure 8: Distribution of the bounding box annotation quality score.

where $N$ is the number of entity categories, $c_i$ is the number of entity with category $i$, $b_i$ is the number of built-in/reliably annotated characters, and $h_i$ is the number of heuristic-annotated characters. Since Tables and figures may not contain any characters but should still be counted as reliable, we set $r_i = 1$ in those cases. We show the distribution of the quality scores in figure 8, where we can see that the distribution is skewed towards documents with higher scores, and a spike at documents with score close to 1.0. For documents with a score close to or exactly 1, we observe that on average they have around $50\%$ fewer text entities, total words and list entities and significantly fewer heading annotations, which is natural since those are the entities that often rely on heuristics rather than built-in styles. Overall, we believe that the annotation quality score reflects the annotation confidence relatively well. However, it is important to note that the highest reliability scores (close to 1) could potentially imply low document diversity, e.g. documents that mainly contain tables and figures as these are always counted with $r = 1.0$.

### A.3 Perplexity Distributions

We provide perplexity distribution plots for the 15 most common languages in figure 9. We observe fairly distinct distributions, with some languages showing more pointed curves (e.g. Ukrainian, Hungarian) and some languages with flat distributions (e.g. Czech, Turkish).

### A.4 Download Failure Statistics

Here we provide further details on reasons that the http requests either failed, or successfully downloaded documents were rejected by WordScape. From table 5, we can see that across snapshots, the most common reason is an unsuccessful http code (e.g. 404), stemming from dead / invalid links found in Common Crawl. This pattern gets amplified as we progress towards older snapshots.

Table 5: Frequency of the most common errors encountered during the download stage. "Other" includes invalid URLs or URLs without a response, exceeded file sizes and miscellaneous errors. "HTTPCode" refers to an unsuccessful HTTP Code (such as 404), "ContentType" an invalid content-type header, "RetryRedirect" too many retries or redirects, "maldoc" a failed OLE check.

|         | Other   | HTTPCode | ContentType | RetryRedirect | Maldoc  | Total Rejections | Checked URLs |
|---------|---------|----------|-------------|---------------|---------|------------------|--------------|
| 2023-14 | 3.847%  | 49.151%  | 11.915%     | 18.121%       | 16.966% | 1,701,770        | 4,142,849    |
| 2023-06 | 5.911%  | 44.832%  | 12.181%     | 18.875%       | 18.200% | 1,616,608        | 3,830,526    |
| 2021-43 | 11.914% | 39.727%  | 9.723%      | 28.329%       | 10.308% | 2,294,023        | 3,400,950    |
| 2020-40 | 0.301%  | 57.873%  | 12.300%     | 18.714%       | 10.812% | 1,718,184        | 2,761,523    |
| 2016-50 | 0.230%  | 52.471%  | 16.524%     | 23.447%       | 7.328%  | 912,971          | 1,209,775    |
| 2013-48 | 0.765%  | 60.941%  | 12.980%     | 20.844%       | 4.469%  | 528,848          | 598,437      |

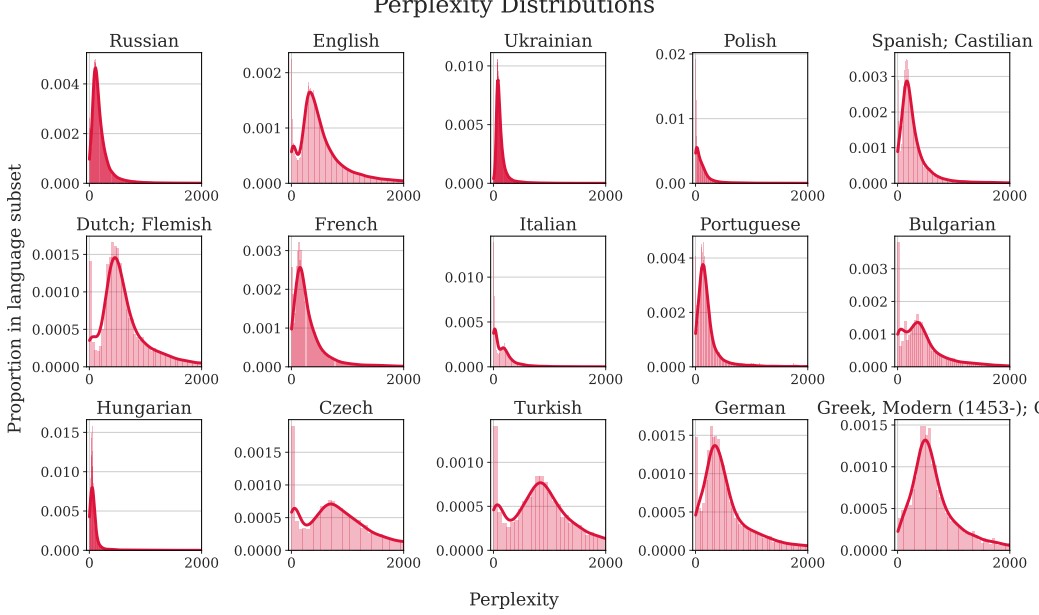

Figure 9: Perplexity Distributions for documents in the Top-15 languages from the November/December 2022 Common Crawl snapshot.

## A.5 Semantic Entity Distributions

Here we provide more details on the semantic entity distributions. The total number of entities for each of the 30 categories annotated by WordScape based on 1.25M documents is presented in table 6. Roughly 50% of all entities are table cells, resulting in a highly imbalanced class distribution. Excluding categories that correspond to elements of a table (i.e. Cells, Rows, Columns), we find a more even distribution with the most dominant entity categories being list items and plain text. Table 7 presents the language specific entity counts, where we have omitted table structure elements and merged the different heading levels into one single category. Figure 10 shows Spearman's rank correlation coefficient for pairs of semantic entity counts at a 5% significance level. Figure 11 indicates how (un)balanced the semantic entity labels are for each language. Specifically, the figure shows the proportion of entities in the top-k semantic entity categories among all entities in the language subset. It can be seen that the Dutch, Russian, Polish and Ukrainian subsets appear to more unbalanced, compared to Spanish or English.

## A.6 Language Specific Topic Modelling

As highlighted in the main part of the paper, there exist considerable differences in the distribution of topics across languages. In Figure 12, it can be seen how the document type diversity varies across languages. While the top-5 categories make up 53.7% of documents, they account for over 80% of Korean and Portuguese documents. Figure 13 provides further evidence for this observation and shows the entire set of top-level categories for each language.

## A.7 Computational Resources

Here we report a detailed breakdown over the computational resources required to process one common crawl snapshot with WordScape. Running the first step of the pipeline, namely parsing of Common Crawl, in a single node setup with 64 CPU cores and 512GB RAM, takes 49 hours to complete, or 3,087 CPU hours. We emphasize that this running time is heavily dependent on the egress speed in the Common Crawl S3 bucket and might vary over time, depending on demand. Running the second step of the pipeline, with 64 cores and 256GB RAM takes 22.5 hours, or 1,440 CPU hours. Finally, the last step of the pipeline is the most CPU intense step and was run on a cluster

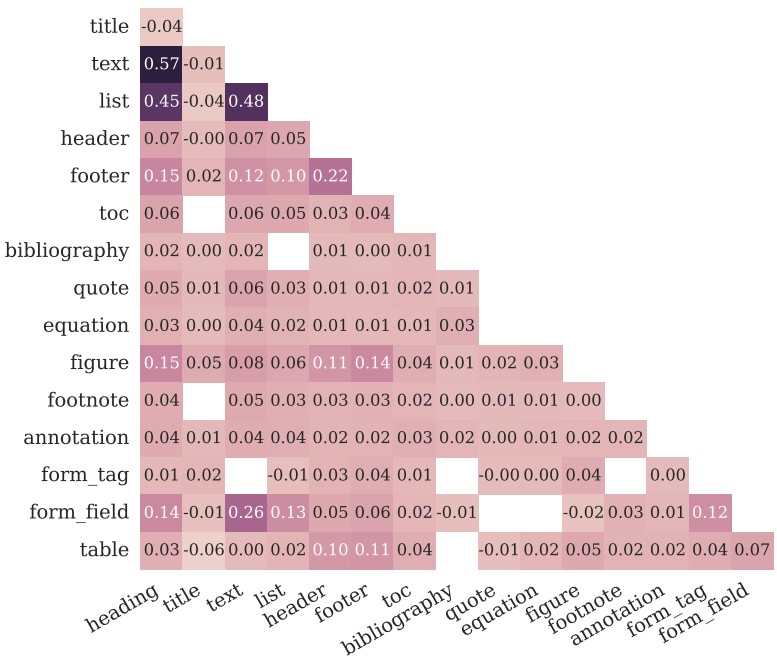

Figure 10: Spearman's rank correlation coefficient for the occurrence of different layout elements. Pairs were the coefficient is not statistically significant at the 5% level are left blank.

| | Top-1 | Top-2 | Top-3 | Top-4 | Top-5 | Top-6 | Top-7 | Top-8 | Top-9 | Top-10 |
|---|---|---|---|---|---|---|---|---|---|---|
| nl | 40.9% | 63.0% | 78.8% | 86.2% | 90.0% | 93.1% | 96.0% | 98.7% | 99.2% | 99.7% |
| ru | 40.8% | 69.5% | 81.2% | 88.9% | 95.9% | 97.2% | 98.4% | 99.6% | 99.9% | 99.9% |
| pl | 37.8% | 59.2% | 73.3% | 87.0% | 91.5% | 94.7% | 97.7% | 99.4% | 99.8% | 99.9% |
| uk | 33.6% | 66.6% | 84.8% | 91.2% | 95.4% | 97.9% | 98.9% | 99.5% | 99.9% | 100.0% |
| fr | 32.9% | 55.2% | 74.0% | 81.1% | 87.1% | 92.9% | 96.3% | 98.8% | 99.5% | 99.7% |
| en | 32.1% | 59.0% | 77.6% | 84.4% | 89.0% | 92.9% | 96.2% | 98.8% | 99.4% | 99.6% |
| es | 30.8% | 56.3% | 73.4% | 79.8% | 85.5% | 90.5% | 95.6% | 98.9% | 99.7% | 99.8% |
| WordScape | 32.0% | 62.3% | 78.0% | 86.0% | 92.0% | 94.7% | 97.0% | 99.2% | 99.7% | 99.8% |

Figure 11: Layout diversity for the top-7 languages. Each column represents the proportion of top-k entities among all semantic entities in the language specific subset. This indicates how (un)balanced the semantic entity labels are for each language. The Dutch, Russian, Polish and Ukrainian subsets appear to more unbalanced, compared to Spanish or English.

with 24 nodes and 24 CPU cores and 96GB RAM each, for about 22 hours, resulting in 12,672 CPU hours. In total, processing one snapshot of Common Crawl thus requires roughly 17k CPU hours.

## A.8 Intended Use

This dataset creation pipeline, and the URLs to Word files extracted from Common Crawl snapshots are intended to be used to generate training data for deep learning based document understanding and language models. The URLs are intended to be processed by the code made publicly available alongside this paper [13]. We emphasize that while the URLs are static and are hosted by the authors, the http responses may change in the course of time. For this reason, we provide SHA-256 hashes of the document contents downloaded during June 1st - 12th 2023, that allow to verify whether the content has changed in any way. Finally, we emphasize that the URLs provided can in some cases point to documents which are protected under copyright law, or otherwise contain sensitive information. It is the responsibility of the user of the URLs to comply with these restrictions.

---

[13] https://github.com/DS3Lab/WordScape

Table 6: Total number of entities after processing 1.25M documents from the November/December 2022 Common Crawl snapshot.

| Entity Category | # Entities | Frequency in Corpus |
|---|---|---|
| Table Cell | 86,995,260 | 0.502776 |
| Table Row | 20,671,174 | 0.119466 |
| List Item | 15,544,374 | 0.089836 |
| Table Column | 15,276,984 | 0.088291 |
| Plain Text | 14,757,476 | 0.085289 |
| Form Field | 3,880,406 | 0.022426 |
| Heading Level 1 | 3,161,658 | 0.018272 |
| Table | 2,935,625 | 0.016966 |
| Heading Level 2 | 1,514,728 | 0.008754 |
| Figure | 1,295,668 | 0.007488 |
| Table Header Cell | 1,273,716 | 0.007361 |
| Footer | 1,145,431 | 0.006620 |
| Heading Level 3 | 1,086,629 | 0.006280 |
| Header | 1,046,474 | 0.006048 |
| Heading Level 4 | 629,615 | 0.003639 |
| Heading Level 5 | 402,500 | 0.002326 |
| Heading Level 6 | 270,680 | 0.001564 |
| Title | 267,039 | 0.001543 |
| Heading Level 9 | 265,021 | 0.001532 |
| Table Header | 170,712 | 0.000987 |
| Heading Level 7 | 166,740 | 0.000964 |
| Heading Level 8 | 119,500 | 0.000691 |
| Table of Contents | 38,479 | 0.000222 |
| Form Tag | 30,110 | 0.000174 |
| Quote | 21,240 | 0.000123 |
| Equation | 18,181 | 0.000105 |
| Table Caption | 16,289 | 0.000094 |
| Footnote | 14,233 | 0.000082 |
| Bibliography | 7735 | 0.000045 |
| Annotation | 6127 | 0.000035 |
| Total | 173,029,804 | 1.000000 |

| | de | en | es | fr | it | ja | ko | nl | pt | ru | zh |
|---|---|---|---|---|---|---|---|---|---|---|---|
| Top-1 | 15.4% | 20.0% | 18.4% | 13.7% | 24.5% | 25.3% | 33.3% | 17.4% | 37.0% | 36.8% | 34.8% |
| Top-2 | 28.5% | 35.4% | 36.5% | 26.9% | 47.1% | 38.1% | 61.1% | 33.8% | 59.2% | 51.7% | 51.5% |
| Top-3 | 38.3% | 46.2% | 50.0% | 38.1% | 58.5% | 49.8% | 72.2% | 45.1% | 72.1% | 61.8% | 59.1% |
| Top-4 | 47.3% | 55.0% | 60.1% | 47.3% | 69.9% | 59.0% | 80.6% | 52.4% | 78.1% | 69.7% | 66.5% |
| Top-5 | 53.7% | 63.1% | 67.1% | 53.7% | 75.2% | 67.2% | 86.1% | 59.0% | 83.6% | 74.5% | 72.5% |

Figure 12: Topic diversity for language based subsets of WordScape data. Each row represents the proportion of documents that are contained in the top-k high level categories obtained via Google CLoud NLP API topic modelling. The French subset is particularly diverse with the top-5 categories making up 53.7% of documents, compared to over 80% for Korean and Portuguese documents.

Table 7: Semantic Entity Counts for each language. Based on 1.25M documents from the Common Crawl November/December 2022 Snapshot.

| lang | total | heading | title | text | list | header | footer | toc | biblio-graphy | quote | equation | figure | footnote | annotation | form tag | form field | table |
|---|---|---|---|---|---|---|---|---|---|---|---|---|---|---|---|---|---|
| ru | 15,775,786 | 1,839,436 | 46,960 | 4,539,836 | 6,431,785 | 211,531 | 180,082 | 7,486 | 65 | 3,554 | 4,313 | 193,374 | 2,266 | 820 | 553 | 1,210,465 | 1,103,260 |
| en | 10,394,549 | 1,935,118 | 67,979 | 3,339,825 | 2,794,441 | 269,703 | 401,096 | 16,373 | 7,408 | 7,181 | 6,767 | 339,312 | 3,416 | 2,167 | 17,485 | 705,912 | 480,366 |
| uk | 4,537,089 | 824,799 | 29,996 | 1,522,893 | 1,499,187 | 45,734 | 18,387 | 1,420 | 0 | 605 | 413 | 109,246 | 186 | 286 | 173 | 191,421 | 292,343 |
| pl | 3,280,527 | 462,993 | 15,714 | 702,104 | 1,240,413 | 55,609 | 95,594 | 1,277 | 0 | 494 | 224 | 105,481 | 1,420 | 627 | 1,278 | 450,001 | 147,298 |
| es | 1,788,687 | 305,786 | 13,305 | 551,226 | 454,999 | 114,504 | 59,704 | 2,720 | 53 | 582 | 554 | 101,980 | 531 | 243 | 1,029 | 90,562 | 90,909 |
| fr | 1,103,862 | 207,277 | 7,754 | 362,840 | 246,549 | 27,710 | 37,202 | 1,705 | 53 | 567 | 503 | 66,105 | 282 | 273 | 2,502 | 78,582 | 63,958 |
| it | 1,063,150 | 160,254 | 5,913 | 304,571 | 193,393 | 21,031 | 24,980 | 468 | 0 | 361 | 56 | 34,210 | 320 | 138 | 440 | 267,528 | 49,487 |
| pt | 877,840 | 166,427 | 5,169 | 263,720 | 178,428 | 44,381 | 31,171 | 280 | 9 | 314 | 236 | 49,188 | 252 | 49 | 946 | 81,461 | 55,809 |
| hu | 823,587 | 148,394 | 2,978 | 292,947 | 236,230 | 21,647 | 17,067 | 256 | 16 | 66 | 37 | 9,583 | 397 | 170 | 126 | 64,923 | 28,750 |
| bg | 740,209 | 158,229 | 3,873 | 207,490 | 211,583 | 17,406 | 22,510 | 131 | 0 | 379 | 27 | 14,652 | 41 | 190 | 75 | 73,610 | 30,013 |
| ja | 736,150 | 119,187 | 16,068 | 385,889 | 61,430 | 10,789 | 15,212 | 235 | 0 | 7 | 49 | 13,215 | 2 | 66 | 860 | 11,147 | 101,994 |
| cs | 694,481 | 139,566 | 6,631 | 196,766 | 215,157 | 10,943 | 16,691 | 289 | 0 | 125 | 299 | 24,642 | 114 | 36 | 329 | 56,004 | 26,889 |
| zh | 593,174 | 114,934 | 9,900 | 204,276 | 110,265 | 15,248 | 28,795 | 864 | 0 | 26 | 305 | 26,803 | 47 | 174 | 57 | 13,350 | 68,130 |
| nl | 578,069 | 91,325 | 2,883 | 236,716 | 127,551 | 42,702 | 15,763 | 216 | 3 | 2,751 | 480 | 18,341 | 535 | 98 | 462 | 21,835 | 16,408 |
| hr | 423,961 | 73,165 | 2,050 | 151,419 | 120,718 | 7,157 | 10,260 | 202 | 0 | 197 | 37 | 10,146 | 90 | 128 | 49 | 25,713 | 22,630 |
| de | 423,375 | 70,926 | 5,453 | 139,918 | 83,645 | 15,871 | 17,986 | 239 | 3 | 143 | 22 | 28,444 | 154 | 64 | 893 | 32,967 | 26,647 |
| tr | 410,355 | 88,572 | 2,009 | 119,679 | 104,771 | 11,264 | 11,925 | 1,296 | 8 | 55 | 37 | 14,321 | 255 | 22 | 177 | 24,697 | 31,267 |
| lv | 380,426 | 52,038 | 1,248 | 81,445 | 185,956 | 5,573 | 9,911 | 145 | 0 | 41 | 3 | 4,201 | 16 | 40 | 74 | 17,781 | 21,954 |
| vi | 377,438 | 65,422 | 1,199 | 83,402 | 136,829 | 5,458 | 8,317 | 377 | 5 | 36 | 277 | 6,828 | 111 | 21 | 65 | 46,125 | 22,966 |
| sk | 367,850 | 63,964 | 1,470 | 137,419 | 89,053 | 6,653 | 13,552 | 65 | 0 | 66 | 98 | 6,545 | 3 | 73 | 368 | 22,744 | 25,777 |
| ro | 329,450 | 55,329 | 1,034 | 80,607 | 83,736 | 5,757 | 9,051 | 19 | 0 | 112 | 11 | 9,843 | 156 | 28 | 14 | 66,021 | 17,732 |
| el | 319,227 | 59,147 | 1,600 | 102,678 | 76,655 | 6,027 | 17,906 | 72 | 2 | 50 | 224 | 13,372 | 163 | 30 | 156 | 18,463 | 22,682 |
| th | 300,554 | 39,930 | 1,158 | 64,980 | 27,600 | 8,047 | 5,215 | 41 | 0 | 23 | 171 | 6,641 | 685 | 7 | 40 | 122,149 | 23,867 |
| ar | 298,431 | 46,699 | 2,456 | 89,117 | 67,563 | 14,320 | 13,841 | 309 | 50 | 237 | 1,145 | 15,685 | 2,185 | 8 | 38 | 23,854 | 20,924 |
| sr | 289,466 | 48,607 | 908 | 94,355 | 72,040 | 4,023 | 9,714 | 106 | 0 | 34 | 29 | 5,351 | 159 | 41 | 155 | 33,144 | 20,800 |
| sl | 214,714 | 34,825 | 1,184 | 67,667 | 66,149 | 4,411 | 5,499 | 96 | 0 | 75 | 17 | 9,519 | 20 | 65 | 76 | 12,532 | 12,579 |
| id | 176,534 | 20,422 | 607 | 67,875 | 44,061 | 6,222 | 8,610 | 211 | 3 | 94 | 423 | 4,482 | 116 | 14 | 50 | 11,707 | 11,637 |
| he | 172,600 | 29,286 | 1,157 | 53,074 | 34,488 | 10,269 | 10,634 | 156 | 41 | 1,810 | 429 | 9,752 | 96 | 39 | 248 | 11,340 | 9,781 |
| sv | 103,126 | 19,525 | 1,107 | 34,003 | 20,422 | 2,272 | 3,314 | 195 | 2 | 267 | 99 | 5,059 | 1 | 26 | 134 | 7,074 | 9,626 |
| no | 91,549 | 17,716 | 1,181 | 29,888 | 22,197 | 2,076 | 2,487 | 50 | 0 | 9 | 14 | 4,320 | 2 | 15 | 218 | 5,731 | 5,645 |
| lt | 87,784 | 10,788 | 514 | 23,136 | 28,844 | 3,785 | 1,062 | 91 | 0 | 2 | 10 | 1,803 | 12 | 1 | 15 | 7,265 | 10,456 |
| fi | 81,001 | 14,818 | 489 | 26,454 | 22,076 | 2,905 | 2,203 | 67 | 3 | 11 | 120 | 3,563 | 1 | 22 | 141 | 3,592 | 4,536 |
| fa | 76,790 | 11,980 | 463 | 21,812 | 13,973 | 2,661 | 2,175 | 349 | 5 | 26 | 46 | 3,206 | 44 | 0 | 268 | 13,455 | 6,327 |
| kk | 60,068 | 9,463 | 140 | 16,191 | 21,014 | 650 | 861 | 15 | 0 | 15 | 2 | 1,696 | 0 | 3 | 0 | 4,799 | 5,219 |
| da | 56,403 | 10,544 | 629 | 18,177 | 14,894 | 1,797 | 2,127 | 20 | 3 | 21 | 24 | 2,508 | 0 | 3 | 38 | 1,762 | 3,856 |
| uz | 55,870 | 6,742 | 35 | 7,681 | 36,317 | 170 | 275 | 0 | 0 | 3 | 20 | 665 | 0 | 0 | 0 | 1,774 | 2,188 |
| ca | 55,217 | 9,480 | 564 | 14,631 | 12,094 | 1,483 | 2,410 | 23 | 0 | 4 | 8 | 4,517 | 1 | 1 | 268 | 6,112 | 3,621 |
| hy | 54,569 | 8,010 | 652 | 18,486 | 16,204 | 325 | 804 | 10 | 0 | 4 | 468 | 980 | 35 | 4 | 1 | 2,486 | 6,100 |
| et | 45,368 | 7,588 | 212 | 10,295 | 19,868 | 498 | 586 | 7 | 0 | 1 | 0 | 722 | 3 | 11 | 24 | 2,967 | 2,586 |
| ko | 40,412 | 6,854 | 680 | 18,080 | 8,458 | 883 | 925 | 0 | 0 | 145 | 0 | 1,220 | 0 | 7 | 0 | 626 | 2,534 |
| ka | 35,052 | 5,985 | 175 | 8,821 | 12,778 | 524 | 853 | 2 | 0 | 8 | 0 | 558 | 2 | 2 | 14 | 641 | 4,689 |
| sq | 31,211 | 8,310 | 94 | 7,923 | 9,627 | 350 | 1,012 | 35 | 0 | 57 | 0 | 890 | 5 | 2 | 0 | 1,059 | 1,847 |
| mk | 26,435 | 4,909 | 152 | 7,712 | 7,135 | 819 | 674 | 1 | 0 | 7 | 0 | 947 | 4 | 9 | 26 | 1,979 | 2,061 |
| ne | 22,267 | 1,527 | 80 | 1,539 | 2,509 | 1,019 | 412 | 3 | 0 | 1 | 1 | 218 | 0 | 0 | 0 | 14,092 | 866 |
| ga | 21,889 | 4,021 | 101 | 4,813 | 7,333 | 745 | 596 | 11 | 0 | 0 | 0 | 1,234 | 0 | 3 | 30 | 946 | 2,056 |
| gl | 19,877 | 2,712 | 72 | 6,292 | 5,696 | 108 | 511 | 4 | 0 | 6 | 0 | 1,490 | 0 | 1 | 5 | 2,052 | 928 |
| cy | 19,538 | 4,938 | 132 | 4,876 | 5,633 | 613 | 1,084 | 371 | 0 | 15 | 0 | 733 | 0 | 95 | 8 | 406 | 634 |
| sh | 16,997 | 3,710 | 48 | 6,831 | 3,478 | 168 | 249 | 0 | 0 | 1 | 9 | 370 | 18 | 1 | 0 | 619 | 1,495 |
| az | 12,951 | 1,731 | 51 | 5,393 | 3,892 | 15 | 26 | 43 | 0 | 0 | 64 | 547 | 6 | 0 | 4 | 353 | 826 |
| is | 10,484 | 1,687 | 103 | 2,988 | 2,135 | 448 | 867 | 0 | 0 | 11 | 0 | 614 | 0 | 1 | 2 | 667 | 961 |
| km | 10,163 | 466 | 14 | 1,078 | 488 | 86 | 16 | 0 | 0 | 0 | 0 | 34 | 0 | 0 | 0 | 7,712 | 269 |
| mn | 10,036 | 1,384 | 31 | 3,480 | 3,319 | 98 | 55 | 0 | 0 | 3 | 58 | 174 | 0 | 0 | 0 | 999 | 435 |
| be | 9,875 | 2,665 | 26 | 3,850 | 2,184 | 307 | 111 | 0 | 0 | 5 | 0 | 58 | 1 | 0 | 25 | 329 | 314 |
| tg | 8,883 | 1,714 | 18 | 2,579 | 3,492 | 18 | 104 | 0 | 0 | 3 | 21 | 196 | 9 | 0 | 0 | 540 | 189 |
| bs | 6,489 | 1,509 | 11 | 2,726 | 1,567 | 64 | 37 | 0 | 0 | 0 | 0 | 69 | 0 | 0 | 0 | 36 | 470 |
| eu | 5,770 | 786 | 24 | 1,459 | 1,641 | 113 | 147 | 15 | 0 | 0 | 1 | 692 | 0 | 0 | 12 | 460 | 420 |
| hi | 5,393 | 1,012 | 39 | 1,501 | 1,276 | 171 | 180 | 19 | 0 | 6 | 0 | 265 | 0 | 0 | 3 | 164 | 757 |
| fy | 5,342 | 752 | 23 | 1,643 | 1,923 | 7 | 179 | 0 | 0 | 4 | 0 | 120 | 0 | 0 | 0 | 625 | 66 |
| bn | 4,932 | 885 | 75 | 1,669 | 273 | 38 | 388 | 2 | 0 | 0 | 0 | 160 | 0 | 0 | 0 | 118 | 1,324 |
| ky | 4,898 | 771 | 6 | 1,710 | 1,408 | 250 | 124 | 0 | 0 | 0 | 0 | 79 | 0 | 0 | 0 | 198 | 352 |
| la | 4,812 | 959 | 82 | 2,969 | 172 | 69 | 76 | 9 | 3 | 3 | 22 | 172 | 0 | 0 | 1 | 42 | 233 |
| mt | 4,664 | 814 | 16 | 1,585 | 1,471 | 6 | 202 | 0 | 0 | 0 | 0 | 74 | 0 | 0 | 0 | 64 | 432 |
| tl | 4,300 | 681 | 19 | 1,027 | 1,101 | 144 | 134 | 0 | 0 | 1 | 0 | 437 | 0 | 0 | 39 | 158 | 559 |
| tt | 4,216 | 1,016 | 21 | 1,325 | 1,477 | 6 | 5 | 12 | 0 | 0 | 0 | 49 | 0 | 0 | 0 | 75 | 230 |
| ur | 3,922 | 752 | 21 | 1,490 | 229 | 173 | 119 | 53 | 0 | 1 | 0 | 746 | 0 | 0 | 0 | 172 | 166 |
| nn | 3,697 | 703 | 47 | 875 | 705 | 179 | 89 | 1 | 0 | 1 | 0 | 171 | 1 | 1 | 59 | 535 | 330 |
| dv | 2,663 | 107 | 32 | 317 | 1,438 | 0 | 209 | 16 | 0 | 0 | 0 | 62 | 0 | 0 | 19 | 101 | 362 |
| ms | 1,973 | 365 | 12 | 561 | 313 | 59 | 49 | 0 | 0 | 0 | 0 | 135 | 0 | 2 | 0 | 362 | 115 |
| mr | 1,968 | 679 | 15 | 618 | 436 | 14 | 7 | 0 | 0 | 0 | 0 | 76 | 0 | 0 | 0 | 0 | 123 |
| sw | 1,783 | 308 | 12 | 693 | 406 | 0 | 2 | 0 | 0 | 3 | 0 | 136 | 46 | 0 | 0 | 170 | 7 |
| mg | 1,637 | 158 | 9 | 479 | 5 | 0 | 369 | 0 | 0 | 611 | 0 | 3 | 0 | 0 | 0 | 0 | 3 |
| eo | 1,558 | 397 | 11 | 596 | 156 | 25 | 18 | 0 | 0 | 0 | 0 | 41 | 0 | 0 | 0 | 101 | 213 |
| lmo | 1,352 | 236 | 2 | 410 | 107 | 46 | 46 | 0 | 0 | 0 | 0 | 423 | 0 | 0 | 0 | 41 | 41 |
| af | 1,152 | 283 | 13 | 379 | 189 | 8 | 5 | 0 | 0 | 0 | 0 | 75 | 0 | 0 | 0 | 31 | 169 |
| krc | 1,070 | 386 | 1 | 266 | 56 | 0 | 0 | 0 | 0 | 0 | 0 | 361 | 0 | 0 | 0 | 0 | 0 |
| am | 997 | 116 | 10 | 175 | 265 | 29 | 91 | 0 | 0 | 0 | 0 | 79 | 0 | 0 | 0 | 65 | 167 |
| ba | 958 | 188 | 6 | 274 | 449 | 0 | 3 | 0 | 0 | 0 | 0 | 0 | 0 | 0 | 0 | 17 | 21 |
| lo | 951 | 166 | 3 | 147 | 538 | 0 | 9 | 0 | 0 | 0 | 0 | 28 | 0 | 0 | 0 | 12 | 48 |
| ps | 830 | 141 | 18 | 190 | 412 | 3 | 6 | 0 | 0 | 0 | 0 | 26 | 0 | 0 | 0 | 15 | 19 |
| pa | 686 | 176 | 15 | 276 | 73 | 0 | 87 | 0 | 0 | 0 | 0 | 48 | 0 | 0 | 0 | 0 | 11 |
| kn | 646 | 269 | 2 | 338 | 22 | 0 | 0 | 0 | 0 | 0 | 0 | 2 | 0 | 0 | 0 | 1 | 2 |
| cv | 571 | 68 | 2 | 186 | 315 | 0 | 39 | 0 | 0 | 0 | 8 | 16 | 0 | 0 | 0 | 81 | 30 |
| my | 531 | 114 | 5 | 152 | 73 | 39 | 13 | 0 | 0 | 0 | 0 | 8 | 0 | 0 | 0 | 0 | 30 |
| ceb | 504 | 1 | 5 | 156 | 6 | 8 | 0 | 0 | 0 | 0 | 0 | 175 | 0 | 0 | 0 | 5 | 148 |
| gu | 471 | 90 | 2 | 127 | 134 | 0 | 51 | 0 | 0 | 0 | 0 | 13 | 0 | 0 | 0 | 34 | 20 |
| rm | 447 | 83 | 3 | 106 | 55 | 33 | 0 | 0 | 0 | 0 | 0 | 3 | 0 | 0 | 38 | 89 | 37 |
| ckb | 438 | 48 | 5 | 129 | 96 | 20 | 5 | 0 | 0 | 0 | 0 | 72 | 0 | 0 | 0 | 1 | 62 |
| yi | 419 | 80 | 3 | 104 | 55 | 4 | 19 | 0 | 0 | 0 | 0 | 23 | 0 | 0 | 0 | 106 | 25 |
| sah | 392 | 87 | 2 | 75 | 145 | 0 | 0 | 0 | 0 | 0 | 0 | 1 | 0 | 0 | 0 | 40 | 42 |
| te | 390 | 116 | 4 | 230 | 4 | 0 | 0 | 0 | 0 | 0 | 0 | 0 | 0 | 0 | 0 | 36 | 0 |
| ta | 341 | 21 | 15 | 188 | 65 | 0 | 27 | 0 | 0 | 0 | 0 | 12 | 0 | 0 | 0 | 0 | 13 |
| si | 315 | 67 | 1 | 101 | 57 | 0 | 57 | 0 | 0 | 0 | 0 | 3 | 0 | 0 | 0 | 28 | 1 |
| ku | 282 | 43 | 1 | 72 | 89 | 0 | 49 | 1 | 0 | 0 | 0 | 1 | 24 | 0 | 0 | 0 | 2 |
| ht | 280 | 33 | 2 | 98 | 53 | 0 | 21 | 0 | 0 | 0 | 0 | 8 | 0 | 0 | 0 | 52 | 13 |
| ml | 271 | 79 | 2 | 117 | 7 | 56 | 0 | 0 | 0 | 0 | 0 | 7 | 0 | 0 | 0 | 0 | 3 |
| jv | 268 | 5 | 0 | 12 | 229 | 10 | 0 | 0 | 0 | 0 | 0 | 3 | 0 | 0 | 0 | 7 | 2 |
| arz | 263 | 28 | 3 | 49 | 53 | 0 | 0 | 0 | 0 | 0 | 0 | 22 | 0 | 0 | 0 | 89 | 19 |
| gd | 238 | 47 | 1 | 108 | 56 | 0 | 0 | 0 | 0 | 0 | 0 | 15 | 0 | 0 | 0 | 0 | 11 |
| ilo | 188 | 1 | 0 | 110 | 61 | 0 | 0 | 0 | 0 | 0 | 0 | 16 | 0 | 0 | 0 | 0 | 0 |
| ia | 128 | 0 | 0 | 0 | 0 | 18 | 15 | 0 | 0 | 0 | 0 | 42 | 0 | 0 | 0 | 0 | 53 |
| oc | 122 | 22 | 1 | 36 | 19 | 0 | 0 | 0 | 0 | 0 | 0 | 8 | 0 | 0 | 0 | 32 | 4 |
| mzn | 102 | 15 | 0 | 32 | 0 | 6 | 0 | 0 | 0 | 0 | 0 | 0 | 0 | 0 | 0 | 47 | 2 |
| war | 96 | 18 | 1 | 18 | 49 | 1 | 0 | 0 | 0 | 0 | 0 | 5 | 0 | 0 | 0 | 0 | 4 |
| lb | 96 | 15 | 2 | 39 | 7 | 0 | 2 | 0 | 0 | 0 | 0 | 1 | 0 | 0 | 0 | 28 | 2 |
| gn | 80 | 4 | 0 | 1 | 75 | 0 | 0 | 0 | 0 | 0 | 0 | 0 | 0 | 0 | 0 | 0 | 0 |
| br | 63 | 0 | 1 | 19 | 0 | 0 | 0 | 0 | 0 | 0 | 0 | 1 | 0 | 0 | 0 | 0 | 42 |
| mhr | 61 | 26 | 0 | 32 | 3 | 0 | 0 | 0 | 0 | 0 | 0 | 0 | 0 | 0 | 0 | 0 | 0 |
| als | 41 | 13 | 1 | 14 | 0 | 0 | 0 | 0 | 0 | 0 | 0 | 0 | 0 | 0 | 0 | 8 | 5 |
| nds | 39 | 0 | 1 | 26 | 0 | 0 | 0 | 0 | 0 | 0 | 0 | 2 | 0 | 0 | 0 | 4 | 6 |
| ug | 33 | 3 | 0 | 10 | 5 | 0 | 0 | 0 | 0 | 0 | 0 | 13 | 0 | 0 | 0 | 0 | 2 |
| new | 14 | 0 | 0 | 0 | 0 | 0 | 0 | 0 | 0 | 0 | 0 | 6 | 0 | 0 | 0 | 0 | 8 |
| bcl | 12 | 0 | 0 | 2 | 0 | 0 | 0 | 0 | 0 | 0 | 0 | 2 | 0 | 0 | 0 | 0 | 8 |
| or | 11 | 1 | 1 | 9 | 0 | 0 | 0 | 0 | 0 | 0 | 0 | 0 | 0 | 0 | 0 | 0 | 0 |
| wuu | 10 | 1 | 1 | 0 | 6 | 0 | 0 | 0 | 0 | 0 | 0 | 0 | 0 | 0 | 0 | 0 | 2 |
| os | 9 | 2 | 0 | 6 | 0 | 0 | 0 | 0 | 0 | 0 | 0 | 1 | 0 | 0 | 0 | 0 | 0 |
| bo | 6 | 1 | 0 | 5 | 0 | 0 | 0 | 0 | 0 | 0 | 0 | 0 | 0 | 0 | 0 | 0 | 0 |
| diq | 2 | 0 | 0 | 0 | 0 | 0 | 0 | 0 | 0 | 0 | 0 | 0 | 0 | 0 | 0 | 0 | 2 |
| kw | 2 | 0 | 0 | 1 | 0 | 0 | 0 | 0 | 0 | 0 | 0 | 0 | 0 | 0 | 0 | 0 | 1 |
| sco | 1 | 0 | 0 | 1 | 0 | 0 | 0 | 0 | 0 | 0 | 0 | 0 | 0 | 0 | 0 | 0 | 0 |

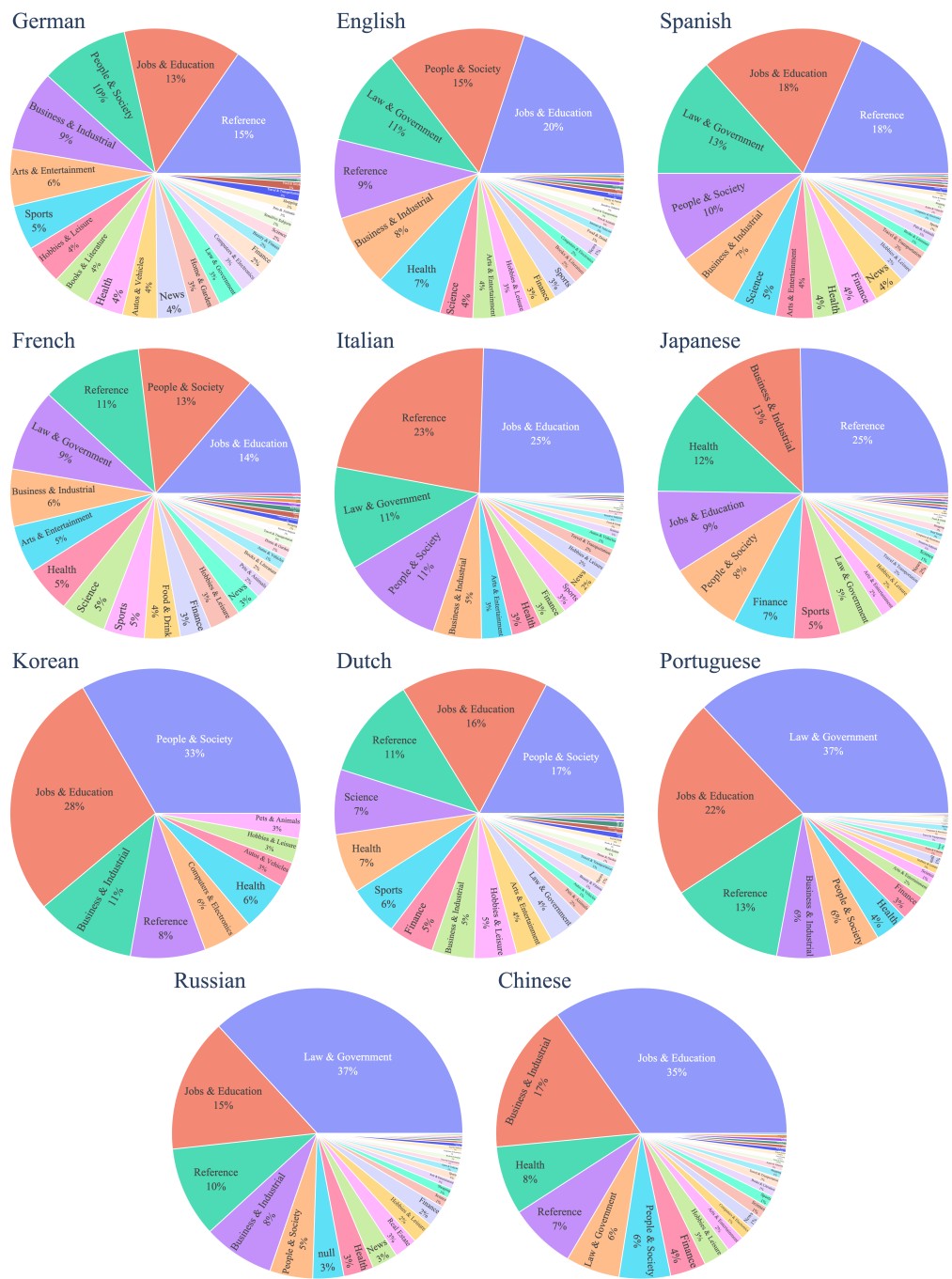

Figure 13: Distribution of top-level content topics discovered via the hierarchical topic classifier from Google Cloud NLP api. The distribution is calculated based on a 25k sample of WordScape data.

