# OpenReview forum: "WordScape: a Pipeline to extract multilingual, visually rich Documents with Layout Annotations from Web Crawl Data"
_NeurIPS.cc/2023/Track/Datasets_and_Benchmarks — NeurIPS 2023 Datasets and Benchmarks Poster_

### Official Review · Reviewer_knqo · 2023-07-21
**A novel pipeline for creating large-scale, multilingual, and multimodal corpus**

**Rating:** 6
**Confidence:** 2

**Strengths:**

* This paper introduces WordScape, the first pipeline for creating a large-scale, multimodal and multilingual corpus for visual document understanding.
* A novel bounding box labeling algorithm based on the Open XML representation is proposed, ensuring high-quality data generation without manual annotation.
* Experimental results validate the efficacy of WordScape by showcasing how pretraining on its datasets significantly alleviates the burden of manual annotation in downstream tasks.
* The authors have open-sourced the code and data, supporting further research in visual document understanding and multimodal models.

**Additional Feedback:**

It would be better to compare existing datasets in a more intuitive manner, such as using tables.

**Clarity:**

This paper provides comprehensive details on data collection and processing, and the overall framework of the paper is clear.

**Correctness:**

Line 240 claims "for Hungarian and Portuguese, the perplexity scores are relatively low." However, in Figure 4, Italian exhibits similarly low perplexity among the seven languages with the lowest number of words. It is also suggested to include some brief explanations about the left figure as well.

**Documentation:**

The author provides details in the paper and supplementary materials on all things about the dataset. However, it appears that the GitHub link provided in the paper is currently invalid.

**Ethics:**

Not Applicable.

**Limitations:**

One limitation discussed in the paper is that bounding box annotation can sometimes produce incorrect results. The authors attribute this to the fact that formatting does not always correlate well with user intent. Possible strategies or plans should be provided.

**Opportunities For Improvement:**

* The paper claims that existing methods for automatically generating labels for document understanding have limitations in terms of domain coverage, language support, and diversity. From my perspective, this seems more like a limitation of the data sources rather than the methods themselves. I am curious about the feasibility of employing these methods to process data from a wide range of domains and languages. A more detailed explanation should be provided to highlight the advantages of the proposed WordScape, such as specific designs tailored to handling diverse data that existing methods cannot address.
* The experiments demonstrate that pretraining on automatically generated corpora can enhance models and reduce the need for annotation in downstream tasks. However, the experiments only compare pretraining on datasets generated from WordScape with no pretraining at all. I suggest adding additional experiments to compare the results of pretraining on other existing datasets. This would further validate the superiority of WordScape. Different experiments can be designed to specifically evaluate its benefits in terms of multi-domain, multi-language, and high-quality data generation.

**Relation To Prior Work:**

This paper discusses related work in Section 2.

**Summary And Contributions:**

This paper presents WordScape, a novel pipeline designed to generate a large-scale, multilingual, and multimodal corpus for document layout detection. The pipeline leverages the Open XML structure of Word documents, which are collected and downloaded from Common Crawl. Meticulous processing techniques are applied to the data, encompassing bounding box annotation, text extraction, language identification, and dataset filters to guarantee the utmost quality. Experimental results demonstrate that pretraining on datasets generated from WordScape can reduce the need for manual annotation in downstream tasks.

---

> ### Author Response · Authors · 2023-08-21
>
> We genuinely thank the reviewer for the comprehensive and succinct summary of our contributions and the constructive feedback provided. Below, we summarize the corresponding changes made in the revision and our answers.
>
> ### Comparison with other methods and pre training datasets
>
> The reviewer is correct in pointing out that the limitations of other large-scale automatically generated datasets stem, to a large extent, from the sources of these datasets. However, the methods developed to annotate these documents are also highly specialized on the source data and are thus not easily extendible to other data sources and hence restricted to specific domains. In addition, related datasets often do not open source the pipeline code. In particular:
>
> - PubLayNet [1] uses documents from PubMed Central and leverages the PMCOA XML structure behind these documents to extract annotations. The technique is thus not easily extendible to other data sources and different data formats.
> - DocBank [2] is based on LaTeX code of documents. The annotation is results from a custom LaTeX parser and is thus not applicable to other sources, for example Word documents.
> - arXivdocs-weak [3] is also based on LaTeX code and has thus equivalent limitations.
>
> We greatly thank the reviewer for encouraging us to more thoroughly present the differences between WordScape and other datasets. In response, in the revision, we have included a new table that contrasts existing corpora and techniques with WordScape.
>
> ### Italian perplexity score
> We greatly thank the reviewer for pointing out that, among the low resource languages, also the italian perplexity scores are among the lowest (in addition to Hungarian and Portugues). We fixed this in the revision and also mentioned Italian.
>
> ### Validity of the Github link in the paper
> The github repo is currently still private, but will be made public after the reviewing process, and after the feedback received has been fully incorporated.
>
> ### References
> [1] Zhong et al. Publaynet: largest dataset ever for document layout analysis. ICDAR, 2019.
>
> [2] Li et al. DocBank: A Benchmark Dataset for Document Layout Analysis. COLING. 2020)
>
> [3] Rausch et al. Docparser: Hierarchical document structure parsing from renderings. AAAI, 2021.

---

> > ### Comment · Reviewer_knqo · 2023-08-25
> >
> > Thank you for the response. It has addressed some of my concerns, but there are still a few remaining:
> >
> > * The lack of experimental comparisons with pre-training on other corporas, as well as the absence of targeted experiments to validate the advantages in multi-domain and multi-language aspects.
> > * The paper does not provide potential solutions to address the limitations of bounding box annotation reliability.
> >
> > Based on these reasons, I decide to maintain my rating.

---

### Official Review · Reviewer_t2Lc · 2023-07-21
**Useful pipeline for extracting and annotating documents from Common Crawl, but with limited contribution**

**Rating:** 6
**Confidence:** 4
**Correctness:** The pipeline construction, analysis a…
**Clarity:** The paper is very clear, and the code…

**Strengths:**

- The dataset generated by the pipeline is intended to contribute a more diverse set of documents along with useful annotations, expanding the semantic scope and content of existing annotated datasets. This may provide a corpus for (pre)training DL models with content of higher quality and coverage (beyond scientific documents). These claims are not supported by the analysis and experiments though.
- The approach (hence pipeline) behind extracting and annotating the documents are well motivated. And the experiments illustrate the utility of the document corpus (besides limitations as explained below).
- The paper is well written; clear and concise, especially the process of running the pipeline and annotating resulting documents are clearly described.

**Additional Feedback:**

- Section 5.2: as mentioned, scientific document corpora exist, so this experiment would benefit from a comparison to e.g. DocBank or ArXiv documents. Minor: it mentions 2,500 manually annotated documents (maybe consider publishing these?) at the start, but a subset of 2,000 are used for fine-tuning, were the other 500 used for evaluation?
- How long does it take to run the annotation (and overall document processing) pipeline? I recommend adding an estimation of the overall duration of running the pipeline start to finish, perhaps in the introduction.

**Documentation:**

The pipeline is well documented in the (yet to be published) repository, with clear READMEs for reusing the pipeline to produce a similar dataset. However, a plan for maintaining this pipeline is not provided, which further limits the value of this contribution as no data is published.

**Ethics:**

No expected concerns, but without a good understanding of the content of the documents, it is hard to judge if there might be ethical consequences in downstream models. As mentioned in my review, I recommend doing more analysis of the actual content besides language (semantically, visually, as well as on the existence of personal identifiable information).

**Limitations:**

The discussion presents limitations, e.g., regarding the annotation quality. It would indeed be very helpful to understand the quality of the annotation component better.

**Opportunities For Improvement:**

- Only the pipeline is released, not the dataset itself. Common Crawl allows redistribution of the (annotated) data, and there is no good motivation provided as to why the documents with annotations are not published. If there is a valid reason for not publishing the data, then at least I expect a plan for maintenance, it should also be made clear what and how long it takes for the full pipeline to run. The current documentation does not provide a convincing plan for persistence and maintenance of the code (it is now published in a personal(?) GitHub repository). I recommend considering Zenodo or another repository for persisting the code. I wonder why the 1.25M documents used in the experiments are not published, or at least the 2,500 hand-annotated documents as used in the experiment in Section 5.2.
- The contribution of the presented pipeline is rather limited the extraction and analysis pipeline are very similar to CCNet (albeit not focused on documents), which is extended with document annotation for which many methods exist. This reduces the novelty of the total pipeline and raises the question about the potential impact of this paper.
- The analysis and experiments do not sufficiently support the claims/motivation of the contribution. The added value of this dataset is not clearly demonstrated against the existing document corpora. For example, one experiment focuses on scientific documents, while the main motivation behind this dataset was multilinguality and cultural/domain diversity, as many large-scale scientific document corpora already exist. It is now unclear what the impact is of the suggested higher quality and coverage? The other two experiments illustrate how the dataset can help reduce the need for labeled datasets but so can the existing large corpora; it is recommended to include a comparison to other large-scale datasets for pretraining to illustrate the added value. Moreover, the analyses fall short with regards to 1) document semantics and quality as suggested as motivation for this source data (what characterizes the content of the documents, beyond the languages), and 2) comparisons with existing datasets.

**Relation To Prior Work:**

There is sufficient coverage of the related work, with distinctions of the presented pipeline highlighted.

**Summary And Contributions:**

**Update 08/25**: after discussion with the authors and the revision of the paper with additional content analysis, comparisons and experiments, I increase my rating from 4 to 6.

---

The paper presents a pipeline, similar to CCNet, for multilingual document extraction from the Common Crawl web corpus. The pipeline is well described, and an analysis is presented that illustrates the scale of the obtained document corpus, the language distributions, and stats of the different entities. Experiments illustrate that a corpus generated with the pipeline can be used to develop pretrained models for document-tasks, such as text detection, and reduce the need for manually annotated documents.

---

> ### Author Response · Authors · 2023-08-21
>
> We genuinely thank the reviewer for the comprehensive and succinct summary of our contributions and the constructive feedback provided. Below, we summarise the corresponding changes made in the revision and our answers.
>
> ### Dataset release
> We do not release the full dataset is the potential violation of copyright issues and privacy concerns. While it is true that Common Crawl allows redistribution of the crawled contents, this is not necessarily the case for the documents embedded as urls in the crawls. We believe that releasing the urls together with the SHA256 hashes of the document content is a fair trade off; on the one hand we do not risk the redistribution of content which their creators did not wish to be redistributed. On the other hand, distributing the hashes of the contents allows for easy verification of integrity of the documents. We will maintain both the code and access to the urls via a public github repository, so that reproducibility is guaranteed as best as possible and researchers can build on the pipeline.
>
> ### Compute requirements
> We report a detailed breakdown of the compute requirements for running the pipeline on a single common crawl snapshot. We break the analysis down into the three main steps of the pipeline, namely 1) parsing of common crawl, 2) downloading of the documents, and 3) annotation. We present these statistics in a separate section in the Appendix.
>
> ### Relation to CCNet / Novelty
> We agree with the reviewer that there are similarities between the CCNet pipeline and the WordScape pipeline. However, we would like to emphasize the following fundamental differences between CCNet and WordScape:
> - Data Source. CCNet processes the plain .wet files generated by common crawl and extracts the text present in the html files. As such, CCNet does not provide any renderings of documents. WordScape, on the other hand, extracts links pointing to word files, and renders them.
> - Labels. WordScape includes a custom MS Word parser for annotating visual elements in the rendered documents, enabling supervised pretraining of multimodal models. CCNet produces unlabelled data for self-supervised training.
> - Multiple Modalities. CCNet produces pure text data. WordScape, in contrast, produces multimodal pairs of text and image data, enabling pretraining of vision models, language models, and multimodal models.
>
> We highly appreciate that the reviewer has encouraged us to present the differences to related work in a more concise manner. We include a more detailed comparison in the revision in table 1.
>
> ### Lack of detailed dataset analysis
> We greatly thank the reviewer for encouraging us to present more detailed analyses of the document layout and the document contents. In the revision, we incorporate this feedback along the following two axes:
> - Layout Analysis. We present an analysis on the distribution of semantic elements in WordScape. In Figure 5, we present the proportions of each semantic element for the overall dataset, and for specific language subsets. We found that all language subsets considered are imbalanced; differences exist in the extent to which the subsets are imbalanced (Fig. 11 in the appendix; e.g. Russian documents are more imbalanced than Spanish documents).
> - Topic modeling. To illustrate the topic diversity in WordScape, we ran the hierarchical topic classifier available in the Google Cloud NLP API on 25k samples. Figure 6 in the revision shows the hierarchy of topics and the diversity present in WordScape. In addition, in the appendix, we show the language-specific distribution of topics, where we again found significant differences between languages.
>
> ### Downstream Tasks
> We agree that the experiments in the initial version of the paper did not sufficiently highlight the contributions of WordScape. To address this, we include additional layout analysis results on DocLayNet. We found that pretraining with WordScape leads to significant improvements for low-resource settings – the benefit is less pronounced in resource-rich settings. However, the number of finetuning samples required to match the performance of WordScape pre-trained models is considerably higher than for other tasks. We attribute this to the observation that layout analysis on DocLayNet is more complex as it comprises more classes and has more diverse layouts. These results are presented in Table 4 in the revision.
>
> ### Clarification on Figure 5
> Figure 5 focuses on performance improvements based on varying pre-training data sizes. We observed significant changes after the 10k pre-train threshold, and beyond steady improvements were seen. This pattern was consistent across models and parameters, leading to two categories: sizes below 10k (2.5k, 5k, 7.5), and above 10k (15k, 20k, 25k). In the revision, we explicitly state these numbers in the caption of Figure 7.
>
> The handcrafted scientific dataset consists of 2.5k manually annotated pages, 2k of which were used for training, and 500 for validation.

---

> > ### Comment · Reviewer_t2Lc · 2023-08-25
> >
> > Thank you for addressing the main concerns from me and other reviewers in the response and revision. I understand the authors' considerations regarding dataset release and respect the decision to not release the raw data (perhaps publishing a smaller and manually curated/checked version might be a valuable contribution in the future). The additional topical and language analysis and comparisons with related work make the paper significantly stronger and underline the contribution of the dataset resulting from the introduced pipeline. I revise my rating accordingly.

---

### Official Review · Reviewer_fwpL · 2023-07-22
**A large-scale document layout dataset is proposed**

**Rating:** 6
**Confidence:** 5
**Clarity:** This paper is well written and easy t…

**Strengths:**

(1) This paper propose a multi-language, cross-disciplinary, and diverse web-scale document image dataset, including layout detection and text content annotations. This is of great help in advancing general document intelligence.

(2) Current automated annotation for large-scale document image datasets is limited in style. This work serves as a supplement to existing data.

(3) The  proposed pipeline can extracts documents in over 100+ languages.

(4). 9.5 million Word doc URLs are released, which can be used to create dataset with 40+ million pages.




**Additional Feedback:**

-  There are a few typos in the main body, such as 'howver' on line 100 and 'section 5' on line 81.

- In the reference, some specific term are not correctly capitalized (e.g., ICDAR, Faster R-CNN); some conference names are abbreviated, while others are not; and many works have already been formally accepted, so the authors should not cite their arXiv preprint versions.

**Correctness:**

The claims made in the submission are correct and the dataset proposed is constructed in a sound way.

**Documentation:**

The authors only mentioned that they would release the URLs of the Word documents but did not specify whether they would make the final dataset annotated through WordScape publicly available. In the submission information, I only saw the shared link for the URLs. Besides, it is unclear whether the Handcrafted Scientific Dataset will be made public.

**Ethics:**

I don’t think there are any ethical concerns.

**Limitations:**

1. Lack of evidence for document type diversity: The major advantage of the proposed method compared to existing approaches is the ability to obtain more diverse data in terms of document types and languages. While Figure 3 presents statistics on language diversity, the paper fails to provide evidence of document type diversity, which is a critical aspect to support the paper's contributions.

2. Insufficient validation of data quality: Annotating categories based on heuristics can lead to significant errors, but authors do not provide a clear analysis of the error rate or its severity. For example, the authors could perform random sampling of the data and compare the results with human annotations to quantify the discrepancies.

3. Pretraining did not lead to the expected performance improvement and even resulted in performance degradation (Tables 1, 2; Figure 5): Considering existing works on document pretraining, large-scale pretraining should typically lead to a stable performance boost in downstream tasks (even when using all downstream task training data). Is this lack of improvement due to poor data quality as mentioned in point 2 above? Would pretraining with other large-scale datasets (e.g., PubLayNet, DocBank) in the same experiment also encounter similar issues? These aspects lack necessary exploration and discussion.

4. The authors mentioned that they would label the 'footer' category, but in the visualization of the first example in Figure 1, the footer is not labeled.

5. In Figure 5, what are the specific number of the pretrained data corresponding to the blue and red lines? Why are they represented as a range? This is confusing, please clarify.


**Opportunities For Improvement:**

1. The advantage of this paper over existing methods is the ability to obtain more diverse data, while the disadvantage is that category annotations are relatively less accurate, which could be improved upon.

2. Since URLs are prone to expiration, the authors should consider to download and organize these Word documents, provide a new access link, and maintain this database. Otherwise, it may lead to subsequent researchers having to perform a lot of repetitive work.


**Relation To Prior Work:**

Yes, this paper clearly discusses how this work differs from previous works.

**Summary And Contributions:**

This paper proposes a pipeline named WordScape for the creation of large-scale document layout detection dataset by parsing the Open XML structure of crawled Word documents. In contrast to the data obtained from existing LaTeX parsing methods, the data obtained by WordScape exhibits a greater variety of styles and languages. This is attributed to the fact that Word documents offer more diversity compared to PDF sources with embedded LaTeX. Authors performed statistical analysis of the obtained dataset and validated the effectiveness of some data through experiments. The URLs of the Word documents and the code for Wordscape are open-source.

---

> ### Author Response · Authors · 2023-08-21
>
> We genuinely thank the reviewer for the comprehensive and succinct summary of our contributions and the constructive feedback provided. Below, we summarize the corresponding changes made in the revision and our answers.
>
> ### Scalability vs. Quality trade-off
>
> The reviewer is correct in pointing out that one of the advantages of the proposed pipeline is that it leads to a diverse corpus of documents, and that the disadvantage is that the annotations are relatively less accurate, compared to human annotations. The reason for this disadvantage is closely linked to an additional advantage of the pipeline, which is its scalability due to the automatically generated annotations (which naturally incur annotation errors). Indeed, while in the paper we present statistics on a 1.25M sample, this number can be scaled much higher if needed (the size of the corpus is limited only by the size of the common crawl database).
>
> ### Dataset release
>
> It is true that some of the published links will expire over time, and that the nature of this dataset is dynamic. The reason that we do not release any downloaded documents publicly, is the potential violation of copyright issues and privacy concerns. We believe that releasing the urls, together with the SHA256 hashes of the document content is a fair trade off; on the one hand we do not risk the redistribution of content which their creators did not wish to be redistributed. On the other hand, distributing the hashes of the contents allows for easy verification of integrity of the documents, guaranteeing that the downloaded document is the same as the document used in the paper. However, we acknowledge that this leads to other researchers having to run step 2 and 3 of the WordScape pipeline as well and that this does not completely eliminate the issue of url expiration.
>
> ### Evidence for document type diversity
>
> We greatly thank the reviewer for encouraging us to present more detailed analyses on the topics present in WordScape documents. To better illustrate the topic diversity of documents in WordScape, we ran the hierarchical topic classifier available in the Google Cloud NLP API [1] on a subset of 25k samples. The full hierarchy of topics is shown in Figure 6 in the revision and highlights the diversity of topics present in WordScape. In addition, in the appendix, we show the language specific distribution of topics, where we found significant differences between language subgroups.
>
> ### Lack of performance boost
>
> We agree with the reviewer that pretraining on WordScape does not lead to performance boosts in all scenarios. This lack of improvements for the benchmark results presented in the initial version of the paper occurs primarily in the settings where the full finetuning dataset is used (e.g. 600 samples for table detection and 2k samples on the handcrafted scientific dataset), while performance boosts appear typically in low resource settings. We hypothesize that the threshold where performance boosts occur is primarily determined by the complexity of the task. To validate this hypothesis, in the revision, we include an additional experiment on the more complex Layout analysis task on DocLayNet. We found that here, significantly more finetuning samples (20k) are required to observe diminishing performance boosts from pretraining. These results are presented in table 4 in the revised manuscript.
>
> Additionally, for the table detection benchmarks, we compute standard deviations and average the mAP scores in order to present less noisy numbers. These updated numbers show that pretraining does not hurt performance.
>
> ### Annotation error in figure 1
>
> The lack of the footer category in the example in Figure 1 is an annotation error (we emphasize that the example was not cherry-picked). This is characteristic of the weak labels generated by the automatic labeling algorithm.
>
> ### Number of pre-training data in Figure 5 [Figure 7 in the revision]
>
> We attempted to communicate the conditional performance improvement across various pre-training data sizes and hyperparameter configurations. We observed a considerable improvement beyond 10’000 pre-train samples. This result was robust across models and hyperparameters.  This lead us to group models in that way (by pre-training sample size below and above this pre-training threshold). The plots and specific confidence bands arose for pre-training samples of 2’500, 5’000 and 7’500 for the <10’000 (red) graphs while 15’000, 20’000, and 25’000 gave rise to the >10’000 (blue) graphs.
>
> ### Typos, Capitalization & References
>
> We greatly thank the reviewer for pointing out these typos, capitalization errors and issues in the references. In the revision, we fixed the typos & capitalization, and updated the references to reflect formally accepted papers.

---

> > ### Comment · Reviewer_fwpL · 2023-08-31
> >
> > I have read the author's rebuttal and appreciate them addressing my concerns. It seems they have provided clarification on the scalability vs quality trade-off, dataset release, evidence for document diversity, and promise to fix some annotation error, typos and reference issues. Since the authors have made a commitment to further improve the paper based on this rebuttal, I will maintain my original rating of marginally above acceptance.

---

### Official Review · Reviewer_qD1P · 2023-07-25
**Overall good work; contribution could be stronger if the full dataset can be released as well as additional dataset analysis/experiments is provided.**

**Rating:** 6
**Confidence:** 3

**Strengths:**

In the era of large (language) models, constructing high-quality datasets at a web scale is increasingly important. While most of the previous work focuses on the generating text-only data, this paper develops a pipeline to produce multimodal layout-rich document data. While there are relevant work before, this paper has the following the strength:
1. The pipeline is carefully designed for processing and extracting MS word documents at web scale (of diverse domains and languages). Similar previous work either focuses on a single (or a few) domains (e.g., scientific documents in PubLayNet or DocBank) and the documents are mostly in the same language. And thus the generated dataset can be a good artifact for pre-training document image understanding models.
2. The pipeline is documented in detail and the processing code will be released (it is still not accessible at the time of review). Most of previous work only releases the compiled dataset and the details of processings are often missing. The detailed documentation of the data processing as well as the released code can be helpful for the document analysis research community to reproduce and develop new document processing pipelines at scale.


**Additional Feedback:**

It would be good if the authors can report the overall CPU hours for processing and constructing the dataset, ideally along with the potential carbon footprint.

**Clarity:**

The paper is well written.


**Correctness:**

- The claim of “At the same time, the model displayed less overfitting to the current task as compared to finetuning on all training data (line 289)” is not supported based on the experimental results. In order to properly test the overfitting / out-of-distribution robustness of layout models, the setup from the recent paper Chen et al. Are Layout-Infused Language Models Robust to Layout Distribution Shifts? A Case Study with Scientific Documents seems to be relevant and proper.


**Documentation:**

The authors properly document the pipeline and the snapshot of the dataset they’ve created; however there is no mention of the plans for maintenance and responsible use.
As word documents might contain PII data, I’d encourage the authors to list the potential concerns of the dataset and clarify the intended and responsible use.


**Limitations:**

The author discussed the limitations of the pipeline including the expirations of the links as well as the potential inaccuracies of bounding box labeling and text extraction.

**Opportunities For Improvement:**

The contribution of this paper can be even stronger if the author can improve along the following axes:
1. Currently the authors do not mention any plans to release the full processed dataset (only links to the word documents will be released, line 77). It is understandable that it is hard to release the full data due to potential copyright issues. However, as the authors acknowledge (line 217), providing only urls might lead to less available files in the future (and thus many accompanying issues like reproducibility). I’d encourage the authors to find a solution to this issue, which can make the contribution of this work much stronger.
2. There is not enough analysis for the visual layouts of the created dataset. One major experiment of this paper is testing the helpfulness of the dataset for different layout analysis benchmarks. However, the authors do not provide any analysis in the main paper (Section 4) in terms of visual layout related aspects of this dataset – e.g., how varied the layout is, what are the types of the documents, or at least additional visualizations of the documents, etc. Including such analysis can make the authors’ claim stronger, and provide additional insights of the potential helpfulness of the dataset in terms of visual layout recognition.
3. The experimental design can be improved.
    - The downstream tasks are not diverse enough. The authors test the pre-trained models on only three English visual layout tasks, while the source data is multi-lingual and is hypothetically more diverse. It would be good to include results on other benchmarks of varied documents and language (considering DocLayNet / PRIMA for different layouts and Europeana Newspapers Project Dataset or the Historical Japanese Documents with Complex Layouts for different languages) to further showcase the helpfulness.
    - Some of the claims are not properly supported, see the following.


**Relation To Prior Work:**

The paper properly contrasts with previous work in this domain.


**Summary And Contributions:**

This paper makes two major contributions:
1. It presents a pipeline that can be used for extracting visual layout structures from MS Word documents at (web) scale.
2. The authors demonstrate that the constructed dataset can be used to pre-train visual layout detectors and it can reduce the manual annotation needed to achieve the same accuracy on the downstream tasks.

---

> ### Author Response · Authors · 2023-08-21
>
> We genuinely thank the reviewer for the thorough and concise summary of our work, its strengths, and the constructive feedback provided. Below, we summarize the corresponding changes made in the revision and our answers.
>
> ### Dataset release
>
> The reviewer is correct in pointing out that the reason that we do not release the documents, is the potential violation of copyright issues and privacy concerns. We believe that releasing the urls, together with the SHA256 hashes of the document content is a fair trade off; on the one hand we do not risk the redistribution of content which their creators did not wish to be redistributed. At the same time, distributing the hashes of the contents allows for easy verification of integrity of the documents, guaranteeing that the downloaded document is the same as the document used in the paper.
>
> ### Analysis on layout and topic diversity
>
> We greatly thank the reviewer for encouraging us to present more detailed analyses of the document layout and topic diversity. In the revision, we incorporate this feedback along the following two axes:
>
> - Layout Analysis. We present an analysis on the distribution of semantic elements in WordScape. In Figure 5, we present the proportions of each semantic element for the overall dataset, as well as for specific language subsets. We found that all language subsets considered are imbalanced, however, there do exist differences in the extent to which the subsets are imbalanced as can be seen from Figure 11 in the appendix (e.g. Russian documents are more imbalanced than Spanish documents).  In the appendix, we also include detailed semantic entity counts for all languages in the dataset.
> - Topic modeling. To illustrate the topic diversity of documents in WordScape, we ran the hierarchical topic classifier available in the Google Cloud NLP API [1] on a subset of 25k samples. The full hierarchy of topics is shown in Figure 6 in the revision and highlights  the diversity of topics present in WordScape. In addition, in the appendix, we show the language specific distribution of topics, where we again found significant differences between language subgroups.
>
> ### Diversity of Downstream Tasks
>
> We agree with the reviewer that the paper can benefit from more diverse benchmark experiments. To address this feedback, we include additional results on the DocLayNet benchmark. We found a similar pattern as in the other benchmarks. Namely, pretraining with WordScape documents significantly improves performance in low-resource settings, while the benefit is less pronounced in resource-rich settings. However, the number of finetuning samples required to match the performance of WordScape pre-trained models is considerably higher compared to the other tasks. We attribute this to the observation that, as opposed to the simpler tasks, layout analysis on DocLayNet is more complex as it comprises more classes and has more diverse layouts. These results are presented in Table 4 in the revision.
>
> ### Correctness of claim “At the same time, the model displayed less overfitting to the current task as compared to fine tuning on all training data (line 289)”
>
> We greatly appreciate the reviewer's comment for pointing out this inaccuracy. We tried to convey the message that pretraining on WordScape leads to a smaller gap between training and testing scores as compared to not using any pretraining data. However, we found this effect negligible overall and removed this sentence in the revision to avoid confusion.
>
> ## Compute requirements for running the WordScape pipeline
>
> We report a detailed breakdown of the compute requirements for running the pipeline on a single common crawl snapshot. We believe this is a valuable contribution as it allows for more accurate resource allocation for researchers. We break the analysis down into the three main steps of the pipeline, namely 1) parsing of common crawl, 2) downloading of the documents, and 3) annotation. We present these statistics in a separate section in the Appendix.
>
> ### Clarify concerns and intended use
>
> We agree with the reviewer in that it is important to emphasize that WordScape can generate datasets that contain samples with potentially copyright-protected content, or may contain otherwise sensitive information. In the revision, we emphasize these concerns clearer in section A.8 on intended use in the appendix.
>
> ### References
> [1] Google Cloud NLP. Google cloud classifying content, https://cloud.google.com/natural-language/docs/classifying-text.

---

### Official Review · Reviewer_nyYp · 2023-07-26
**A large dataset for visual document understanding**

**Rating:** 8
**Confidence:** 4
**Correctness:** Claims are correct
**Clarity:** The paper is well written.

**Strengths:**

- A large dataset for visual document understanding from the web.
- The process for dataset construction is described in great detail, which is helpful for other researchers to reproduce the process.
- The result shows the benefits of pretraning models using this dataset, which can reduce the labeling cost on the target domain.


**Additional Feedback:**

n/a

**Documentation:**

The paper provides sufficient details

**Ethics:**

The dataset may contain contents that are unsafe, improper, subject to copyright restrictions. But only pipeline and urls will be distributed.


**Limitations:**

Malware is detected. But unsafe contents and copyright issues don’t seem to be addressed.

**Opportunities For Improvement:**

- No empirical results show the value of multilingual resources.
- Pretraining doesn’t lead to better performance on resource-rich settings, indicating that the bottleneck of document understanding may not be the close-domain pretraining data. More analyses are welcomed.


**Relation To Prior Work:**

Prior Work is clearly discussed

**Summary And Contributions:**

This study introduces a large-scale dataset for visual document understanding. The dataset is constructed by crawling and processing Word documents on the internet, resulting in 9.5M documents and over 40M pages. Automatic annotation is applied to produce weak labels (bounding boxes) for training detection models. The results demonstrate that this dataset can be utilized for pretraining a base document understanding model, reducing the amount of annotation on target domains.

---

> ### Author Response · Authors · 2023-08-21
>
> We genuinely thank the reviewer for the comprehensive and succinct summary of our contributions and the constructive feedback provided. Below, we summarize the corresponding changes made in the revision and our answers.
>
> ### Value of multilingual resources
>
> We agree with the reviewer in that the value of multilingual resources can be further explored. Indeed, developing multilingual, multimodal document understanding models is a challenging research problem in its own right. We view WordScape as an important step in this direction, and believe that the development of such models is exciting future research. In the revised paper, we have highlighted this in the discussion.
>
> ### Performance on resource-rich settings
>
> We agree with the reviewer that - as more high-quality labeled data is available - pretraining on WordScape provides a diminished performance improvement. However, we believe that the amount of high-quality labeled data required increases with the task's complexity. In the initial version of the paper, the considered benchmark tasks were relatively simple such that even with 500 - 1000 manually labeled samples, pretraining did not yield significant performance improvements. In the revision, we include additional results on the more complex DocLayNet Benchmark. We found that considerably more finetuning samples are needed to close the gap between pretraining and no pretraining.
>
> We believe that the value of the scalability of WordScape will become even more apparent as large-scale experiments with higher capacity models which also take into account language (e.g. LayoutLM [1], Ernie-Layout [2]) are conducted.
>
> ### Malware and Copyright Issues
>
> We detect malware vulnerabilities in a very conservative way. For example, we discard any Word document containing macros or which is otherwise flagged as potentially unsafe by OLETools. Nevertheless, this still does not guarantee safety and could be exploited by motivated adversaries. In particular, an adversary can buy a dead domain present in our database and construct a malicious docx file. We emphasize this issue more prominently in Section 3.2 in the revision.
>
> We address copyright issues by explicitly not redistributing downloaded word files as these can potentially include personally identifiable information (PII), or might otherwise be protected by intellectual property rights. It is thus the duty of the user to use the generated data in a manner that complies with copyright law. In the revision, we have emphasized this point in the intended use section in the appendix.
>
> [1] Huang et al. Layoutlmv3: Pre-training for document ai with unified text and image masking. Proceedings of the 30th ACM International Conference on Multimedia. 2022.
> [2] Qiming et al. Ernie-layout: Layout knowledge enhanced pre-training for visually-rich document understanding. arXiv preprint arXiv:2210.06155. 2022.

---

### Author Response · Authors · 2023-08-21
**Summary**

We sincerely thank all the reviewers for their constructive feedback and suggestions. We believe that all reviewers brought essential points to our attention, which we aim to address comprehensively. In this response, we want to summarize the central aspects of the revision and how we address each issue. In the revised paper, we highlight additional content added in response to the reviewing process in blue font color.

### Document Type diversity
While we have shown document diversity in terms of the languages in the document corpora generated with WordScape, several reviewers highlighted that the paper lacks an analysis of the types of documents. To address this issue, we conducted a detailed document type analysis using the google cloud NLP api [1] for content classification. Specifically, we extracted the text from a random sample of 25’194 documents and classified it into one of 1’091 categories. In the revision, we conduct a detailed analysis in the new section 5 and present our findings for the overall topic distribution, language-specific topic distributions, and document type diversity.

### Layout Diversity
Several reviewers encouraged us to present a more fine-grained analysis of the diversity of the layout annotations in addition to the overall semantic entity category counts shown in the appendix. We highly appreciate this suggestion and conducted a more detailed analysis. In particular, we present the language-specific and overall distribution of layout elements and a correlation analysis on counts of different pairs of layout elements in the revision.

### Dataset Release
We are releasing WordScape not as a fixed set of document images and their annotations but as a list of urls and the SHA-256 hashes of the document contents. While reviewers encouraged us to publish the documents as a static dataset, we explicitly refrain from such a step for the following reasons.
- Some of these documents might fall under copyright protection and may contain sensitive information such as credit card numbers, social security numbers, or other personally identifiable information.
- The nature of the contribution is in itself not static but rather a dynamic pipeline that can be used to create datasets. This is a key advantage over other works in the same area, as it allows dynamically creating new datasets as time progresses. The importance of using up-to-date pretraining data has recently been emphasized in [2].
- We believe that releasing the urls, together with the SHA256 hashes of the document content is a fair trade-off; on the one hand, we do not risk the redistribution of content which their creators did not wish to be redistributed. At the same time, distributing the hashes of the contents fosters reproducibility as it allows for easy verification of integrity of the documents, guaranteeing that the downloaded document is the same as the document used in the paper.

### Benchmark Evaluations
Several reviewers pointed out that the benchmark experiments in the initial version of the paper could be more diverse and give an incomplete picture of the value of WordScape. We welcome these suggestions and, to address these issues, present results in the experiments section on the more complex DocLayNet benchmark for layout analysis. We discovered a similar trend as in the other benchmark results: pretraining on WordScape yields the most significant performance boost in lower resource settings. Nevertheless, the high-quality fine-tuning samples required to match performance with the pre-trained model is considerably higher in this setting. We attribute this observation to the increased complexity of the layout analysis task.

### Comparison with existing Document Corpora
Several reviewers encouraged us to provide further comparisons with other pretraining document corpora. We are glad about these suggestions as this will help researchers better understand the overall landscape and the strengths and weaknesses of WordScape. In the revision, we now include a table in the related work section that lists attributes of the core datasets in the field.

### References
[1] Google Cloud NLP. Google cloud classifying content, https://cloud.google.co m/natural-language/docs/classifying-text.

[2] Longpre, Shayne, et al. "A Pretrainer's Guide to Training Data: Measuring the Effects of Data Age, Domain Coverage, Quality, & Toxicity." arXiv preprint arXiv:2305.13169 (2023).

---

### Decision · Program_Chairs · 2023-09-22

**Decision:**

Accept (Poster)

**Comment:**

Reviewers agree this is an okay paper, although have several concerns and suggestions. Many of these are addressed in the responses & revision.

Summary And Contributions:
Presents a pipeline "WordScrape" that parses Open XML in Word documents from Common Crawl. This pipeline generates a multilingual, multimodal corpus for document layout detection. The corpus is described and experiments suggest that it reduces the need for manually annotated documents.

Strengths:
- Offers a large, diverse, multimodal dataset.
- The pipeline to create the dataset is the first of its kind.
- The pipeline is well-described.
- Proposes a novel bounding box labeling algorithm.
- Useful for pretraining models, alleviating some of the need for manual annotation in downstream tasks.
- The authors are open-sourcing the code and data.

Opportunities For Improvement and Limitations:
- Only the pipeline is released, not the dataset itself.
- The dataset is composed of links, not the actual documents, yet handling the fact that links expire is not adequately addressed.
- Authors could better explain the specific contribution of WordScrape with respect to data quality, diversity, and coverage.
- The claim that this approach in particular reduces the need for manual annotation is not fully supported: the experiments only compare pretraining on datasets generated from WordScape with no pretraining at all. (Authors respond that the revision now includes more information on this.)
- More analysis analysis of the visual layouts of the created dataset.
- Experiments can be improved (more specific suggestions provided by reviewers).
- The quality of the annotations can be better documented.

Correctness:
Addressed in author response/revision.

Clarity:
The paper is well written.

Relation To Prior Work:
Prior work is clearly discussed.

Documentation:
What's provided is done well. Most issues are addressed in response and revision. Remaining issue seems to be that authors don't specify plans for maintenance, and could be more detailed in suggestions for responsible use of the dataset.